# TabRepo: A Large Scale Repository of Tabular Model Evaluations and its AutoML Applications

## Abstract

We introduce TabRepo, a new dataset of tabular model evaluations and predictions. TabRepo contains the predictions and metrics of 1206 models evaluated on 200 classification and regression datasets. We illustrate the benefit of our dataset in multiple ways. First, we show that it allows to perform analysis such as comparing Hyperparameter Optimization against current AutoML systems while also considering ensembling at marginal cost by using precomputed model predictions. Second, we show that our dataset can be readily leveraged to perform transfer-learning. In particular, we show that applying standard transfer-learning techniques allows to outperform current state-of-the-art tabular systems in accuracy, runtime and latency.

## 1 Introduction

Machine learning on structured tabular data has a long history due to its wide range of practical applications. Significant progress has been achieved through improving supervised learning models, with key method landmarks including SVM (Hearst et al., 1998), Random Forest (Breiman, 2001) and Gradient Boosted Trees (Friedman, 2001). The performance of base models is still being improved by a steady stream of research, for instance using new paradigms such as pretraining of transformer models (Hollmann et al., 2022) or combining non-parametric and deep-learning methods (Gorishniy et al., 2023) which also improves the performance of downstream AutoML systems (Gijsbers et al., 2022; He et al., 2021).

AutoML solutions were shown to perform best in the large scale benchmarks performed by (Erickson et al., 2020; Gijsbers et al., 2022). Auto-Sklearn (Feurer et al., 2015a; 2020) was an early approach that proposed to select pipelines to ensemble from the Sklearn library and meta-learn the hyperparameter-optimization (HPO) with offline evaluations. The approach was successful and won several AutoML competitions. Several frameworks followed with other AutoML approaches such as TPOT (Olson & Moore, 2016), H2O AutoML (LeDell & Poirier, 2020), and AutoGluon (Erickson et al., 2020). AutoGluon particularly showed strong performance by combining ensembling (Caruana et al., 2004), stacking (Wolpert, 1992) and bagging (Breiman, 1996). While all techniques were shown to be important to reach good accuracy, they also bear a significant cost in terms of training time as models are fitted on several folds of the training data and the stacking of models strongly impacts inference latency.

The proliferation of AutoML and supervised learning methods led to several works focusing on benchmarking tabular methods. Recently, Gijsbers et al. (2022) proposed a unified benchmark called the AutoMLBenchmark to compare tabular methods. However, the cost of running such comparisons for new methods becomes quickly prohibitive. Evaluating a single method in the AutoMLBenchmark requires 40000 CPU hours of compute [1]. This limits the number of methods present in the benchmark and restricts research and experimentation to those with access to sufficient computational resources. For instance, measuring the impact of ensembling requires retraining the base models which can easily become too expensive in particular given many datasets and seeds.

---

[*]Equal contribution

[1]The CPU hour requirement is based on running the full 104 datasets in AutoMLBenchmark across 10 folds for both 1 hour and 4 hour time limits on an 8 CPU machine.

To address this issue, we introduce TabRepo , a dataset of model evaluations and predictions. The main contributions of this paper are:

- A large scale evaluation of tabular models comprising 723600 model predictions with 1206 models from 6 different families which are evaluated across 200 datasets and 3 seeds.
- We show how the repository can be used to study at marginal cost the performance of tuning models while considering ensembles by leveraging precomputed model predictions.
- We show that our dataset combined with transfer learning achieves a result competitive with state-of-the-art AutoML systems and outperforms others by a significant amount in accuracy and training time.
- We study the performance of transfer learning techniques on tabular methods across several novel angles such as data efficiency, training time, and prediction latency.

This paper first reviews related work before describing the TabRepo dataset. We then illustrate how TabRepo can be leveraged to compare HPO with ensemble against current state-of-the-art tabular systems and finally show how transfer-learning can be used to outperform current systems.

## 2 RELATED WORK

Acquiring and re-using offline evaluations to eliminate redundant computation has been proposed in multiple compute intensive fields of machine learning. In HPO, several works proposed to acquire a large number of evaluations to simulate the performance of different optimizers across many seeds which can easily become prohibitive otherwise, in particular when the blackbox function optimized involves training a large neural network (Klein & Hutter, 2019; Eggensperger et al., 2021). Similarly, tabular benchmarks were acquired for Neural Architecture Search (Ying et al., 2019; Dong & Yang, 2020) as it was observed that, due to the large cost of comparisons, not enough seeds were used to distinguish methods properly from random-search (Yang et al., 2020).

While the cost of tabular methods can be orders of magnitude lower than training large neural networks, it can still be significant in particular when considering many methods, datasets, and seeds. Several works proposed to provide benchmarks with precomputed results, in particular Gorishniy et al. (2021) and Grinsztajn et al. (2022). One key differentiator with those works is that our work exposes model *predictions* and *prediction probabilities* which enables to simulate instantaneously not only the errors of single models but also *ensembles* of any subset of available models. To the best of our knowledge, the only prior works that considered providing a dataset compatible with ensemble predictions is Borchert et al. (2022) in the context of time-series and Purucker & Beel (2022) in the context of tabular prediction. Our work differs from Purucker & Beel (2022) in several ways. First, they consider 31 classification datasets whereas we include 200 datasets both from regression and classification. Also, they only considered base models whereas our dataset contains AutoML system evaluations that allows to compare different strategies with state-of-the-art systems. Finally, another limitation is that different models were evaluated on each dataset, making it hard to learn fixed portfolios or model selections strategies and simulate their performance on a holdout dataset without the use of imputation.

Another important advantage of acquiring offline evaluations is that it allows to perform transfer-learning, e.g. to make use of the offline data to speed up the tuning of model hyperparameters. In particular, a popular transfer-learning approach is called Portfolio learning, or Zeroshot HPO, and consists in selecting greedily a set of models that are complementary and are then likely to perform well on a new dataset (Xu et al., 2010). Due to its performance and simplicity, the method has been applied in a wide range of applications ranging from HPO (Wistuba et al., 2015), time-series (Borchert et al., 2022), computer vision (Arango et al., 2023), tabular deep-learning (Zimmer et al., 2021), and AutoML (Feurer et al., 2015a; 2020).

The current state-of-the-art for tabular predictions in terms of accuracy is arguably AutoGluon (Erickson et al., 2020) in light of recent large scale benchmarks (Gijsbers et al., 2022). The method trains models from different families with bagging: each model is trained on several distinct non-overlapping random splits of the training dataset to generate out-of-fold predictions whose scores are likely to align well with performance on the test set. Then, another layer of models is trained whose inputs are both the original inputs concatenated with the predictions of the models in the

previous layers. Finally, an ensemble is built on top of the last layer model predictions using ensemble selection (Caruana et al., 2004). Interestingly, this work showed that excellent performance could be achieved without performing HPO and instead using a fixed list of manually selected model configurations. However, the obtained model can be expensive for inference due to the use of model stacking and requires human experts to select default model configurations. Our work shows that using TabRepo, one can alleviate both caveats by learning default configurations which improves accuracy and latency when matching compute budget.

## 3   TABREPO

We now describe TabRepo and our notations to define its set of evaluations and predictions. In what follows, we denote $[n] = \{1, \ldots, n\}$ to be the set of the first $n$ integers.

**Model bagging.**   All models are trained with *bagging* to better estimate their hold-out performance and improve their accuracy. Given a dataset split into a training set $(X^{(\text{train})}, y^{(\text{train})})$ and a test set $(X^{(\text{test})}, y^{(\text{test})})$ and a model $f^{\lambda}$ with parameters $\lambda$, we train $\mathcal{B}$ models on $\mathcal{B}$ non-overlapping cross-validation splits of the training set denoted $\{(X^{(\text{train})}[b], y^{(\text{train})}[b]), (X^{(\text{val})}[b], y^{(\text{val})}[b])\}_{b=1}^{\mathcal{B}}$. Each of the $\mathcal{B}$ model parameters are fitted by ERM, i.e. by minimizing the loss

$$\lambda_b = \arg\min_{\lambda} \mathcal{L}(f^{\lambda}(X^{(\text{train})}[b]), y^{(\text{train})}[b]), \quad \text{for } b \in [\mathcal{B}].$$

where the loss $\mathcal{L}$ is calculated via root mean-squared error (RMSE) for regression, the area under the receiver operating characteristic curve (AUC) for binary classification and log loss for multi-class classification. We choose these evaluation metrics to be consistent with the AutoMLBenchmark defaults (Gijsbers et al., 2022).

One can then construct *out-of-fold predictions*[2] denoted as $\tilde{y}^{(\text{train})}$ that are computed on unseen data for each bagged model, i.e. predictions are obtained by applying the model on the validation set of each split i.e. $f^{\lambda_b}(X^{(\text{val})}[b])$ which allows to estimate the performance on the training set for unseen data. To predict on a test dataset $X^{(\text{test})}$, we average the predictions of the $\mathcal{B}$ fitted models,

$$\tilde{y}^{(\text{test})} = \frac{1}{\mathcal{B}} \sum_{b=1}^{\mathcal{B}} f^{\lambda_b}(X^{(\text{test})}). \tag{1}$$

**Datasets, predictions and evaluations.**   We collect evaluations on $\mathcal{D} = 200$ datasets from OpenML (Vanschoren et al., 2014). For selecting the datasets, we combined two prior tabular dataset suites. The first is from the AutoMLBenchmark (Gijsbers et al., 2022), and the second is from the Auto-Sklearn 2 paper (Feurer et al., 2020). Refer to Appendix C for a detailed description of the datasets.

For each dataset, we generate $\mathcal{S} = 3$ tasks by selecting the first three of ten cross-validation fold as defined in OpenML's evaluation procedure, resulting in $\mathcal{T} = \mathcal{D} \times \mathcal{S}$ tasks in total. The list of $\mathcal{T}$ tasks' features and labels are denoted

$$\{((X_i^{(\text{train})}, y_i^{(\text{train})}), (X_i^{(\text{test})}, y_i^{(\text{test})}))\}_{i=1}^{\mathcal{T}}$$

where $X_i^s \in \mathbb{R}^{\mathcal{N}_i^s \times d_i}$ and $y_i \in \mathbb{R}^{\mathcal{N}_i^s \times o_i}$ for each split $s \in \{\text{train}, \text{test}\}$, $\mathcal{N}_i^s$ denotes the number of rows available in each split. Feature and label dimensions are denoted with $d_i$ and $o_i$ respectively. We use a loss $\mathcal{L}_i$ for each task depending on its type, in particular we use AUC for binary classification, log loss for multi-class classification and RMSE for regression.

For each task, we fit each model on $\mathcal{B} = 8$ cross-validation splits before generating predictions with Eq. 1. The predictions on the training and test splits for any task $i \in [\mathcal{T}]$ and model $j \in [\mathcal{M}]$ are denoted as

$$\tilde{y}_{ij}^{(\text{train})} \in \mathbb{R}^{\mathcal{N}_i^{(\text{train})} \times o_i}, \qquad \tilde{y}_{ij}^{(\text{test})} \in \mathbb{R}^{\mathcal{N}_i^{(\text{test})} \times o_i}. \tag{2}$$

We can then obtain losses for all tasks and models with

$$\ell_{ij}^{(\text{train})} = \mathcal{L}_i(\tilde{y}_{ij}^{(\text{train})}, y_i^{(\text{train})}), \qquad \ell_{ij}^{(\text{test})} = \mathcal{L}_i(\tilde{y}_{ij}^{(\text{test})}, y_i^{(\text{test})}). \tag{3}$$

For all tasks and models, we use the AutoGluon featurizer to preprocess the raw data prior to fitting the models (Erickson et al., 2020).

---

[2]Note that for classification tasks, we refer to *prediction probabilities* as simply *predictions* for convenience.

**Models available.**   For base models, we consider RandomForest (Breiman, 2001), ExtraTrees (Geurts et al., 2006), XGBoost (Chen & Guestrin, 2016), LightGBM (Ke et al., 2017), CatBoost (Prokhorenkova et al., 2018), and Multi-layer perceptron (MLP) [3] We evaluate all *default* configurations used by AutoGluon for those base models together with 200 random configurations for each family yielding $\mathcal{M} = 1206$ configurations in total. All configurations are run for one hour. For the models that are not finished in one hour, we early stop them and use the best checkpoint according to the validation score to generate predictions.

In addition, we evaluate 6 AutoML frameworks: Auto-Sklearn 1 and 2 (Feurer et al., 2015a; 2020), FLAML (Wang et al., 2021), LightAutoML (Vakhrushev et al., 2021), H2O AutoML (LeDell & Poirier, 2020) and AutoGluon (Erickson et al., 2020). AutoGluon is evaluated for the three presets "medium", "high" and "best" and all frameworks are evaluated for both 1h and 4h fitting time budget. We run all model configurations and AutoML frameworks via the AutoMLBenchmark (Gijsbers et al., 2022), using the implementations provided by the AutoML system authors.

For every task and model combination, we store losses defined in Eq. 3 and predictions defined in Eq. 2. Storing evaluations for every ensemble would be clearly infeasible given the large set of base models considered. However, given that we also store base model predictions, an ensemble can be fit and evaluated on the fly for any set of configurations by querying lookup tables as we will now describe.

**Ensembling.**   Given the predictions from a set of models on a given task, we build ensembles by using the Caruana et al. (2004) approach [4] The procedure selects models by iteratively picking the model such that the average of selected models' predictions minimizes the error. Formally, given $\mathcal{M}$ model predictions $\{\tilde{y}_1, \ldots, \tilde{y}_{\mathcal{M}}\} \in \mathbb{R}^{\mathcal{M}}$, the strategy selects $\mathcal{C}$ models $j_1, ..., j_{\mathcal{C}}$ iteratively as follows

$$j_1 = \arg\min_{j_1 \in \mathcal{M}} \mathcal{L}(\tilde{y}_{j_1}, y^{(\text{train})}), \qquad j_n = \arg\min_{j_n \in \mathcal{M}} \mathcal{L}(\frac{1}{n}\sum_{c=1}^{n} \tilde{y}_{j_c}, y^{(\text{train})}).$$

The final predictions are obtained by averaging the selected models $j_1, \ldots, j_{\mathcal{C}}$:

$$\frac{1}{\mathcal{C}}\sum_{c=1}^{\mathcal{C}} \tilde{y}_{j_c}. \tag{4}$$

Note that the sum is performed over the vector of model indices which allow to potentially select a model multiple times and justifies the term "weight". In practice, the number of selected models $\mathcal{C}$ is selected by early-stopping, i.e. by adding models as long as the validation error decreases.

Critically, the performance of any ensemble of configurations can be calculated by summing the predictions of base models obtained from lookup tables. This is particularly fast as it does not require any retraining but only recomputing losses between weighted predictions and target labels.

## 4   COMPARING HPO AND AUTOML SYSTEMS

We now show how TabRepo can be used to analyze the performance of base model families and the effect of tuning hyperparameters with ensembling against recent AutoML systems. All experiments are done at marginal costs given that they just require querying precomputed evaluations and predictions.

### 4.1   MODEL ERROR AND RUNTIME DISTRIBUTIONS

In Fig. 1, we start by analyzing the performance of different base models. In particular, the rank of model losses over datasets shows that while some model families dominate in performance on

---

[3]TabRepo also contains other families of models such as K-Nearest-Neighbors, TabPFN and FT-transformer (Gorishniy et al., 2021). Due to these models not running successfully for all tasks and some requiring GPU or pretraining, we run our main evaluations without them and share the results with those models in appendix F.

[4]We consider only simple ensembling methods since our goal is to illustrate how TabRepo can be leveraged to evaluate state-of-the-art systems, see (Purucker & Beel, 2023) for ensembling methods that can outperform (Caruana et al., 2004).

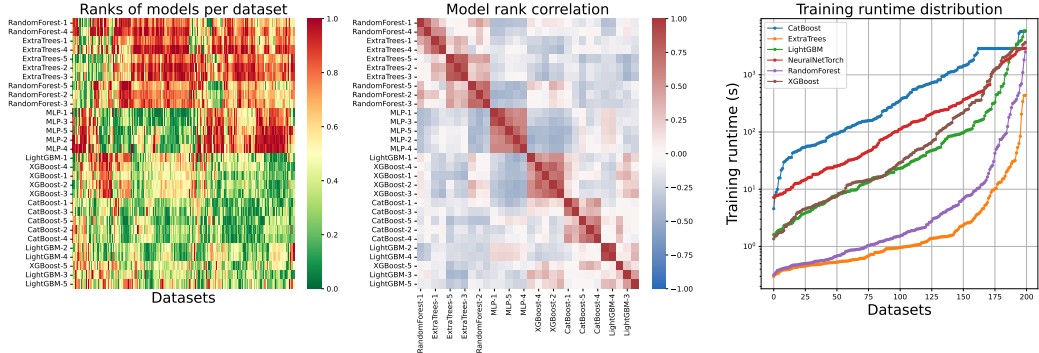

Figure 1: Cluster map of model rank for all datasets (left), correlation of model ranks (middle) and average runtime distribution over every dataset (right). For readability, only the first 5 configurations of each model family are displayed in the left and middle figures.

aggregate such as gradient boosted methods CatBoost and LightGBM, in some tasks MLP are better suited. Looking at model correlations, we see interesting patterns as some model families are negatively correlated between each other such as MLP and XGBoost which hints at the potential benefit of ensembling.

Next, we plot the distribution of runtime configurations over all 600 tasks. We see that an order of magnitude separates respectively the training runtime of CatBoost from MLP, XGBoost and LightGBM, with the remaining methods being faster still. Importantly, while CatBoost obtains the strongest average rank among model families, it is also the most expensive which is an important aspect to take into account when considering possible training runtime constraints as we will see later in our experiments.

## 4.2 Effect of tuning and ensembling on model error

We now compare methods across all tasks by using both ranks and normalized errors. Ranks are computed over the $\mathcal{M}$ different models and all AutoML frameworks. Normalized errors are computed by reporting the relative distance to a topline loss compared to a baseline with

$$\frac{l_{\text{method}} - l_{\text{topline}}}{l_{\text{baseline}} - l_{\text{topline}}}$$

while clipping the denominator to 1e-5 and the final score value to $[0, 1]$. We use respectively the top and median score among all scores to set the topline and baseline. The median allows to avoid having scores collapse when one model loss becomes very high which can happen frequently for regression cases in presence of overfitting or numerical instabilities.

**Comparison.**   In Fig. 2 and Tab. 1, we show respectively the whole distribution and the aggregate of our two metrics across all tasks.

For each model family, we evaluate the default hyperparameter, the best hyperparameter obtained after a random search of 4 hours and an ensemble built on top of the best 20 configurations obtained by this search. As previously seen in Fig. 1, CatBoost dominates other models and LightGBM is the runner-up.

In Fig. 2, we see that tuning model hyperparameters improves all models while ensembling allows LightGBM to match CatBoost. No model is able to beat state-of-the-art AutoML systems even with tuning and ensembling. This is unsurprising as all state-of-the-art tabular methods considered multiple model families in order to reach good performance and echoes the finding of Erickson et al. (2020).

Table 1: Normalized-error, rank, training and inference time averaged over all tasks given 4h training budget. Inference time is calculated as the prediction time on the test data divided by the number of rows in the test data.

| method | normalized-error | rank | time fit (s) | time infer (s) |
|---|---|---|---|---|
| Portfolio (ensemble) | 0.394 | 172.0 | 6715.5 | 0.050 |
| AutoGluon best | 0.406 | 203.6 | 5565.3 | 0.062 |
| Portfolio | 0.462 | 230.7 | 6715.3 | 0.012 |
| Autosklearn2 | 0.476 | 238.6 | 14415.9 | 0.013 |
| AutoGluon high | 0.482 | 276.6 | 5435.3 | 0.002 |
| Lightautoml | 0.490 | 240.8 | 9188.0 | 0.298 |
| Flaml | 0.531 | 310.1 | 14269.8 | 0.002 |
| H2oautoml | 0.544 | 329.9 | 13920.0 | 0.002 |
| AutoGluon medium | 0.549 | 304.7 | 367.7 | 0.001 |
| CatBoost (tuned + ensemble) | 0.557 | 260.6 | 9120.8 | 0.011 |
| LightGBM (tuned + ensemble) | 0.559 | 257.5 | 3507.5 | 0.009 |
| CatBoost (tuned) | 0.562 | 272.9 | 9124.4 | 0.002 |
| LightGBM (tuned) | 0.591 | 294.6 | 3527.2 | 0.001 |
| MLP (tuned + ensemble) | 0.610 | 394.5 | 5781.3 | 0.101 |
| CatBoost (default) | 0.614 | 332.4 | 443.7 | 0.002 |
| MLP (tuned) | 0.646 | 441.1 | 5775.5 | 0.014 |
| XGBoost (tuned + ensemble) | 0.657 | 346.7 | 4973.8 | 0.013 |
| XGBoost (tuned) | 0.670 | 368.4 | 4964.7 | 0.002 |
| LightGBM (default) | 0.747 | 478.7 | 54.2 | 0.001 |
| XGBoost (default) | 0.768 | 509.4 | 73.2 | 0.002 |
| MLP (default) | 0.782 | 611.3 | 39.7 | 0.015 |
| ExtraTrees (tuned + ensemble) | 0.800 | 526.1 | 597.4 | 0.001 |
| ExtraTrees (tuned) | 0.818 | 553.5 | 597.6 | 0.000 |
| RandomForest (tuned + ensemble) | 0.819 | 558.7 | 1507.9 | 0.001 |
| RandomForest (tuned) | 0.830 | 575.8 | 1507.3 | 0.000 |
| ExtraTrees (default) | 0.889 | 762.3 | 3.8 | 0.000 |
| RandomForest (default) | 0.896 | 749.4 | 17.5 | 0.000 |

Table 2: Win rate comparison for 4 hour time limit with the same methodology as Erickson et al. (2020). Win rate is computed against a portfolio ensemble (ties count as 0.5). The re-scaled loss is calculated by setting the best solution to 0 and the worst solution to 1 on each dataset, and then normalizing and taking the mean across all datasets. Rank, fit time, and infer time are averaged over all tasks.

| method | winrate | > | < | = | time fit (s) | time infer (s) | loss (rescaled) | rank |
|---|---|---|---|---|---|---|---|---|
| Portfolio (ensemble) (4h) | **0.500** | | | 200 | 6722.4 | 0.050 | **0.253** | **3.192** |
| AutoGluon best (4h) | 0.465 | 91 | 105 | 4 | 5565.3 | 0.062 | 0.287 | 3.433 |
| Autosklearn2 (4h) | 0.378 | 74 | 123 | 3 | 14415.9 | 0.013 | 0.395 | 4.330 |
| Lightautoml (4h) | 0.270 | 52 | 144 | 4 | 9188.0 | 0.298 | 0.429 | 4.638 |
| CatBoost (tuned + ensemble) (4h) | 0.235 | 46 | 152 | 2 | 9128.3 | 0.009 | 0.508 | 4.995 |
| Autosklearn (4h) | 0.302 | 59 | 138 | 3 | 14413.6 | 0.009 | 0.509 | 5.053 |
| Flaml (4h) | 0.310 | 60 | 136 | 4 | 14269.8 | 0.002 | 0.530 | 5.055 |
| H2oautoml (4h) | 0.233 | 45 | 152 | 3 | 13920.0 | 0.002 | 0.555 | 5.305 |

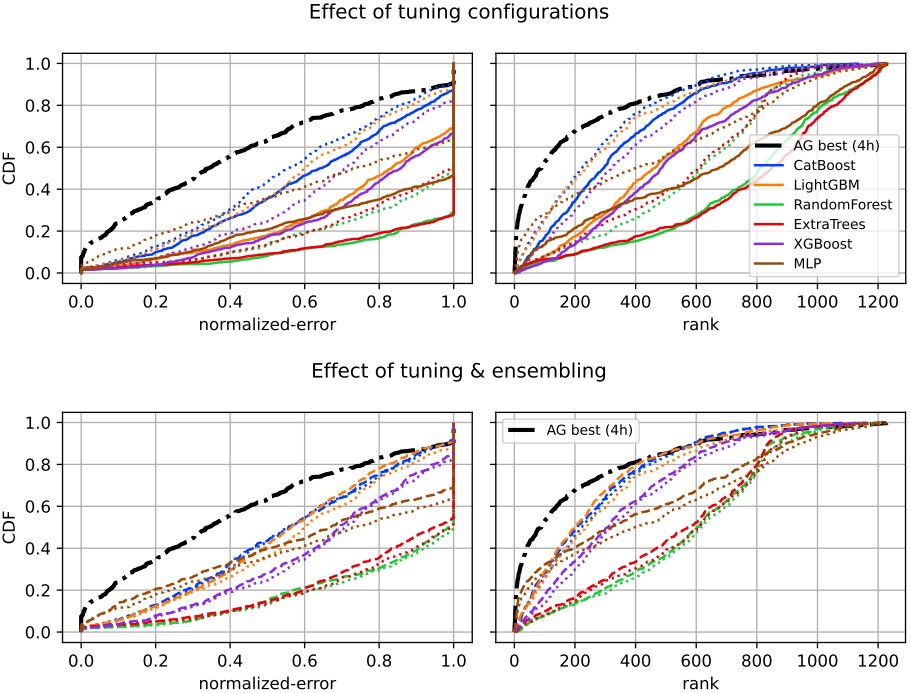

Figure 2: Cumulative distribution function of normalized-errors (left) and ranks (right) for all model families. The line-style denotes respectively the performance of the default configuration (top, solid), of the best configuration after 4h of tuning (top and bottom, dotted) and of an ensemble built on top of the best tuned configurations for the same budget (bottom, dashed).

## 5 PORTFOLIO LEARNING WITH TABREPO

In the previous section, we saw how TabRepo can be leveraged to analyze the performance of frameworks when performing tuning and ensembling. In particular, we saw that ensembling a model family after tuning does not outperform current AutoML systems. We now show how TabRepo can be combined with transfer learning techniques to perform the tuning search offline and outperform current AutoML methods.

**Portfolio learning.** To leverage offline data and speed-up model selection, Xu et al. (2010) proposed an approach to learn a portfolio of complementary configurations that performs well on average when evaluating all the configurations of the portfolio and selecting the best one.

Similarly to Caruana ensemble selection described in Eq. 4, the method iteratively selects $\mathcal{N} < \mathcal{M}$ configurations as follows

$$j_1 = \underset{j_1 \in [\mathcal{M}]}{\arg\min} \, \mathbb{E}_{i \sim [\mathcal{T}]}[\ell_{ij_1}^{(\text{train})}], \qquad j_n = \underset{j_n \in [\mathcal{M}]}{\arg\min} \, \mathbb{E}_{i \sim [\mathcal{T}]}[\underset{k \in [n]}{\min} \, \ell_{ij_k}^{(\text{train})}].$$

At each iteration, the method greedily picks the configuration that has the lowest average error when combined with previously selected portfolio configuration.

**Anytime portfolio.** Fitting portfolio configurations can be done in an *any-time* fashion given a fitting time budget. To do so, we evaluate portfolio configurations sequentially until the budget is exhausted and use only models trained up to this point to select an ensemble. In cases where the first configuration selected by the portfolio takes longer to run than the constraint, we instead report the result of a fast baseline as in Gijsbers et al. (2019).

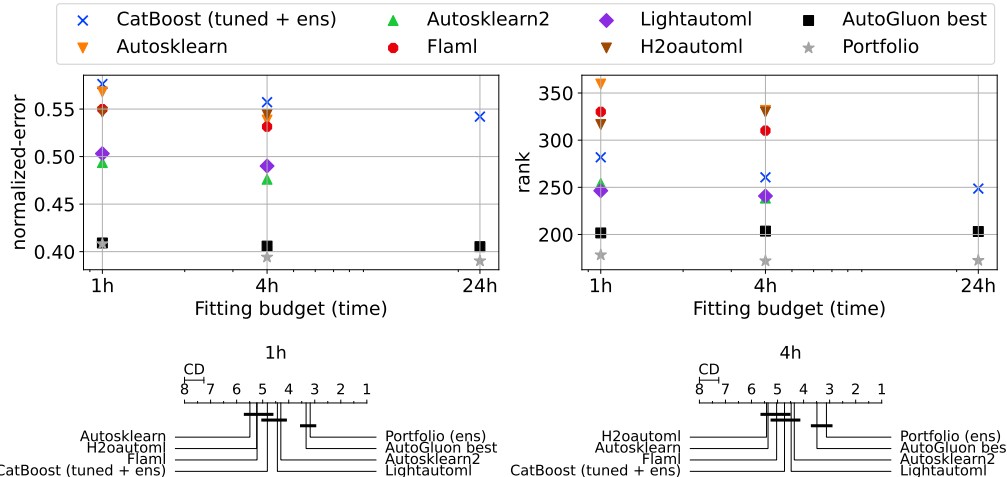

Figure 3: Top: scatter plot of average normalized error (left) and rank (right) against fitting training time budget. Bottom: Critical difference (CD) diagram showing average rank between method selected and which methods are tied statistically by a horizontal bar.

**Evaluations.** We evaluate the anytime portfolio approach in a standard leave-one-out setting. When evaluating on the $i$-th dataset, we compute portfolio configurations on $\mathcal{D} - 1$ training datasets by excluding the $i$-th test dataset to avoid potential leakage.

Results are reported in Tab. 1 when considering a 4h fitting budget constraint. We report both the performance of the best model according to validation error ("Portfolio") and when ensembling the selected portfolio configurations ("Portfolio (ensemble)"). The portfolio combined with ensembling outperforms AutoGluon for accuracy and latency given the same 4h fitting budget even without stacking. When only picking the best model without ensembling, the portfolio still retains good performance and outperforms all frameworks other than AutoGluon while having a very low latency. We also report win rate following the methodology of Erickson et al. (2020) in Tab. 2 which confirms the same result, namely the portfolio obtained from TabRepo outperforms other AutoML methods.

In Fig. 3, we report the performance for different fitting budgets. Ensembles of portfolio configurations can beat all AutoML frameworks for all metrics for 1h, 4h and 24h budget without requiring stacking which allows to obtain a lower latency compared to AutoGluon. Critical difference (CD) diagrams from Demšar (2006) show that while portfolio has better aggregate performance than other methods, AutoGluon and Portfolio are tied statistically. Those two methods are the only methods that are statistically better than all baselines. Interestingly among AutoML systems besides AutoGluon, only AutoSklearn 2 and LightAutoML are better than a baseline consisting of tuning and ensembling CatBoost models although the methods are tied statistically to this baseline.

As in the previous section, all evaluations are obtained from pre-computed results in TabRepo. This demonstrates another potential use of TabRepo, namely to be able to design a system combining transfer learning and ensembling that can reach state-of-the-art performance and compare against a wide variety of methods at marginal compute cost.

**How much data is needed?** We have seen that TabRepo allows to learn portfolio configurations that can outperform state-of-the-art AutoML systems. Next, we analyze the question of how much data is needed for transfer learning to achieve strong results in two dimensions, namely: how many offline configurations and datasets are required to reach good performance? While important, these dimensions are rarely analyzed in previous transfer learning studies due to their significant cost, however they can be obtained in a cheap fashion with TabRepo.

In Fig. 4, we vary both of those dimensions independently. When evaluating on a test dataset, we pick a random subset of configurations $\mathcal{M}'$ per model family in the first case and a random subset of $\mathcal{D}' < \mathcal{D}$ datasets in the second case and report mean and standard error over 10 different seeds.

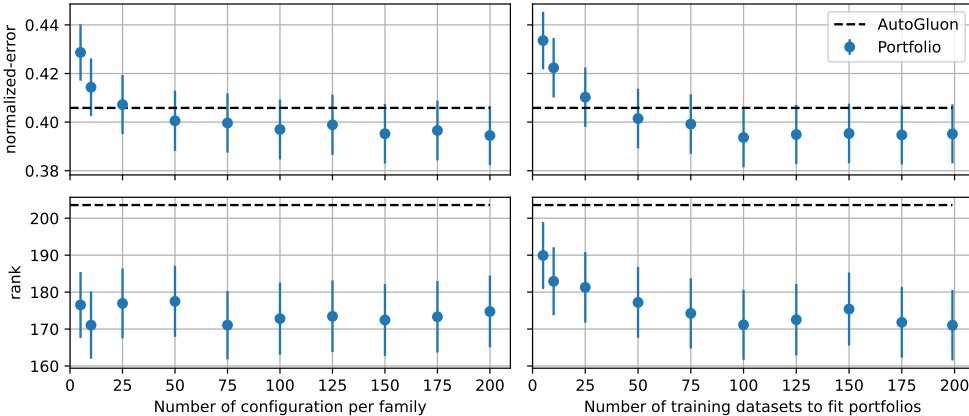

Figure 4: Effect of number of configuration per family (left) and number of training dataset (right) on normalized-error (top) and rank (bottom). All methods are fitted under a 4h fitting budget.

Portfolio with ensembling starts outperforming AutoGluon at around 50 configurations or datasets. Having more datasets or more configurations in offline data both improve the final performance up to a certain point with a saturating effect around 100 offline configurations or offline datasets.

## 6 LIMITATIONS

**Cost.** Evaluating offline configurations is expensive. In total, 26592 hours on a m6i.2xlarge instance on AWS were needed to complete all model evaluations of TabRepo which translates to 212736 CPU hours. However, performing the analysis done in this paper without leveraging precomputed evaluations and predictions would have costed 86415 hours on a m6i.2xlarge which translates to 691320 CPU hours which is $\sim 3.2$ times more expensive. We hope that the repository can be used to test more research ideas which would further amortize its cost.

**Dataset features.** While previous works were able to demonstrate improvements when taking dataset features (Feurer et al., 2015b; Jomaa et al., 2021), we were not able to obtain similar improvement over simple portfolio methods. We postulate this may be due to a need of human feature engineering or it may also be that the large number of datasets used to learn the portfolios makes conditioning on dataset features less critical as seen in (Feurer et al., 2020).

**Transformers.** We did not include transformer models e.g. (Gorishniy et al., 2021) as their training cost can be significantly higher and their performance against other tabular methods such as Gradient Boosted Trees is still being investigated (Grinsztajn et al., 2022).

## 7 CONCLUSION

In this paper, we introduced TabRepo, a benchmark of tabular models on a large number of datasets. Critically, the repository contains not only model evaluations but also predictions which allows to efficiently evaluate ensemble strategies. We showed that the benchmark can be used to analyze the performance of different tuning strategies combined with ensembling at marginal cost. We also showed how the dataset can be used to learn portfolio configurations that outperforms state-of-the-art tabular methods for accuracy, training time and latency.

The code for accessing evaluations from TabRepo and evaluating any ensemble will be made available with the camera ready together with the scripts used to generate all the paper results. We hope this paper will facilitate future research on new methods combining ideas from CASH, multi-fidelity and transfer-learning to further improve the state-of-the-art in tabular predictions.

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

## A  ADDITIONAL EXPERIMENT DETAILS

**Number of Caruana Steps.**  In all our experiments, we set the number of Caruana steps to $\mathcal{C} = 40$ when building ensembles of base models or portfolio configurations. We observe that values beyond 40 provide negligible benefit while linearly increasing the runtime of simulations in TabRepo.

**Fallback method.**  We use the default configuration of Extra-trees as a backup when the first configuration of a portfolio does not finish under the constraint which takes just a few seconds to evaluate.

**Number of portfolio configurations.**  When reporting results on a portfolio, we apply the anytime procedure described in Sec 5 and run at most $\mathcal{N} = 200$ portfolio configurations. Setting this bound serves mostly as an upper-bound as all configurations are almost never evaluated given that all configurations have to be trained under the fitting budget. We investigate the effect of increasing the number of portfolio configurations in Fig. 5.

**Hardware details.**  All model configuration and AutoML framework results were obtained on AWS EC2 machines via AutoMLBenchmark's AWS mode functionality. For all model and AutoML evaluations, we used m6i.2xlarge EC2 instances with 100 GB of gp3 storage. These instances have 8 virtual CPUs (4 physical CPUs) and 32 GB of memory. The Python version used for all experiments was 3.9.18. We chose m6i.2xlarge instances to align with the AutoMLBenchmark's choice of m5.2xlarge instances. m5 instances have the same number of CPUs and memory, but the m6i instances were more cost-efficient due to faster CPUs.

All simulation paper experiments in Sec. 4 and 5 were done on a m6i.32xlarge to avoid memory issues and take less than one hour.

**Critical difference diagrams.**  We use Autorank (Herbold, 2020) to compute critical difference diagrams.

**Data-structure.**  TabRepo takes 107 GB on disk. To avoid requiring large memory cost, we use a memmap data-structure which loads model evaluations on the fly from disk to memory when needed. This allows to reduce the RAM requirement to  20GB of RAM.

## B  API

In Listing 1, we show an example of calling TabRepo to retrieve the ensemble performance of a list of models. Because we store all model predictions, we are able to reconstruct the metrics of any ensemble among the $\mathcal{M} = 1206$ models considered.

```
1  from tabrepo import EvaluationRepository
2
3  # load TabRepo with 200 datasets, 3 folds and 1416 configurations
4  repository = EvaluationRepository.from_context(version="D244_F3_C1416_200
       ")
5
6  # returns in ~2s the tensor of metrics for each dataset/fold obtained
       after ensembling the given configurations
7  metrics = repository.evaluate_ensemble(
8      datasets=["balance-scale", "page-blocks"],  # dataset to report
       results on
9      folds=[0, 1, 2],  # which folds to consider for each dataset
10     configs=["CatBoost_r42_BAG_L1", "NeuralNetTorch_r40_BAG_L1"],  #
       configs that are ensembled
11     ensemble_size=40,  # maximum number of Caruana steps
12 )
13
14 # returns the predictions on the val data for a given task and config
15 val_predictions = repository.predict_val(
16     dataset="page-blocks", fold=2, config="ExtraTrees_r7_BAG_L1"
17 )
18
19 # returns the predictions on the test data for a given task and config
20 test_predictions = repository.predict_test(
21     dataset="page-blocks", fold=2, config="ExtraTrees_r7_BAG_L1"
22 )
```

Listing 1: Example of calling TabRepo to obtain performance scores on an ensemble configuration or model predictions on validation/test splits.

## C  DATASET DETAILS

For selecting the datasets, we combined two prior tabular dataset suites. The first is from the AutoMLBenchmark (Gijsbers et al., 2022), containing 104 datasets. The second is from the Auto-Sklearn 2 paper (Feurer et al., 2020), containing 208 datasets.

All datasets are publicly available via OpenML. After de-duplicating, the union contains 289 datasets. The AutoMLBenchmark datasets have been previously filtered from a larger set via a specific inclusion criteria detailed in section 5.1.1 of Gijsbers et al. (2022). Notably, they filter out datasets that are trivial, such that simple methods such as a random forest cannot perfectly solve them. We perform a similar operation by fitting a default Random Forest configuration on all 289 datasets and filtering any dataset that is trivial (AUC $> 0.999$, log loss $< 0.001$, or r2 $> 0.999$). After filtering trivial datasets, we are left with 244 datasets.

We then run all AutoML baselines and model configurations on the 244 datasets (3 folds, for a total of 732 tasks). We performed re-runs on failed tasks when necessary to attempt to get results on all models and AutoML systems for every dataset, but sometimes this was not possible due to problems such as out-of-memory errors or AutoML system implementation errors outside our control. For datasets with model or AutoML system failures, we exclude them. We exclude datasets rather than impute missing values to ensure the results being computed are fully verifiable and replicable in practice. After excluding datasets with model or AutoML system failures, we have 211 datasets remaining.

Finally, we filter out the 11 largest datasets for practical usability purposes of TabRepo. This is because loading the prediction probabilities of 1206 model configurations on large (multi-class) datasets leads to significant challenges. As an example, the total size of the predictions for 211 datasets is 455 GB. By reducing to 200 datasets, the size decreases dramatically to 107 GB (The full 244 datasets is 4.5 TB).

In total, we use 105 binary classification datasets, 67 multi-class classification datasets and 28 regression datasets. We provide a table of dataset summary statistics in Tab. 3 and an exhaustive list of the 200 datasets used in TabRepo separated by problem type in Tab. 4, Tab. 5 and Tab. 6 where

Table 3: Statistics of the 200 datasets in TabRepo

|  | n | f |
|---|---|---|
| mean | 17722 | 570 |
| std | 55270 | 2161 |
| min | 100 | 3 |
| 5% | 500 | 3 |
| 10% | 575 | 5 |
| 25% | 1107 | 10 |
| 50% | 3800 | 20 |
| 75% | 10027 | 60 |
| 90% | 41173 | 506 |
| 95% | 73134 | 1787 |
| max | 583250 | 10936 |

we list for each dataset the TaskID OpenML identifier, the dataset name, the number of rows $n$, the number of features $f$ and the number of classes $C$ which is always 2 for binary classification.

### C.1 TRAIN-TEST SPLITS

For all datasets we use the OpenML 10-fold Cross-validation estimation procedure and select the first 3 folds for our experiments. For each task (a particular dataset fold), we use 90% of the data as training and 10% as test. We use identical splits to Gijsbers et al. (2022).

## D MODEL DETAILS

For each model type, we used the latest available package versions when possible. The precise versions used for each model are documented in Tab. 7.

For each model family, we choose 201 configurations, 1 being the default hyperparameters, as well as 200 randomly selected hyperparameter configs.

The search spaces used are based on the search spaces defined in AutoGluon. We expanded the search range of various hyperparameters for increased model variety. Note that selecting the appropriate search space is a complex problem, and is not the focus of this work. TabRepo is built to work with arbitrary model configurations, and we welcome the research community to improve upon our initial baselines.

For all models we re-use the AutoGluon implementation for data pre-processing, initial hyperparameters, training, and prediction. We do this because choosing the appropriate pre-processing logic for an individual model is complex and introduces a myriad of design questions and potential pitfalls.

For maximum training epochs / iterations, instead of searching for an optimal value directly, we instead rely on the early stopping logic implemented in AutoGluon which sets the iterations to 10,000 for gradient boosting models and epochs to 500 for MLP.

### D.1 MODEL CONFIG FULL RESULTS

Refer to the supplementary file results_ranked_configs.csv for the complete set of results for all model configs on all tasks. This file includes the training runtime, inference time (per row), and metric error for all models on each task.

### D.2 MODEL CONFIG HYPERPARAMETERS

Refer to the supplementary files located in the folder configs/ for the hyperparameters used for each model config.

Table 4: Binary classification datasets used in TabRepo.

| Task ID | name | n | f | C | Task ID | name | n | f | C |
|---|---|---|---|---|---|---|---|---|---|
| 3593 | 2dplanes | 40768 | 10 | 2 | 3783 | fri_c2_500_50 | 500 | 50 | 2 |
| 168868 | APSFailure | 76000 | 170 | 2 | 3606 | fri_c3_1000_10 | 1000 | 10 | 2 |
| 359979 | Amazon_employee_acce | 32769 | 9 | 2 | 3581 | fri_c3_1000_25 | 1000 | 25 | 2 |
| 146818 | Australian | 690 | 14 | 2 | 3799 | fri_c3_500_10 | 500 | 10 | 2 |
| 359967 | Bioresponse | 3751 | 1776 | 2 | 3800 | fri_c3_500_50 | 500 | 50 | 2 |
| 359992 | Click_prediction_sma | 39948 | 11 | 2 | 3608 | fri_c4_500_100 | 500 | 100 | 2 |
| 361331 | GAMETES_Epistasis_2- | 1600 | 1000 | 2 | 3764 | fried | 40768 | 10 | 2 |
| 361332 | GAMETES_Epistasis_2- | 1600 | 20 | 2 | 189922 | gina | 3153 | 970 | 2 |
| 361333 | GAMETES_Epistasis_2- | 1600 | 20 | 2 | 9970 | hill-valley | 1212 | 100 | 2 |
| 361334 | GAMETES_Epistasis_3- | 1600 | 20 | 2 | 3892 | hiva_agnostic | 4229 | 1617 | 2 |
| 361335 | GAMETES_Heterogeneit | 1600 | 20 | 2 | 3688 | houses | 20640 | 8 | 2 |
| 361336 | GAMETES_Heterogeneit | 1600 | 20 | 2 | 9971 | ilpd | 583 | 10 | 2 |
| 359966 | Internet-Advertiseme | 3279 | 1558 | 2 | 168911 | jasmine | 2984 | 144 | 2 |
| 359990 | MiniBooNE | 130064 | 50 | 2 | 3904 | jm1 | 10885 | 21 | 2 |
| 3995 | OVA_Colon | 1545 | 10935 | 2 | 359962 | kc1 | 2109 | 21 | 2 |
| 3976 | OVA_Endometrium | 1545 | 10935 | 2 | 3913 | kc2 | 522 | 21 | 2 |
| 3968 | OVA_Kidney | 1545 | 10935 | 2 | 3704 | kdd_el_nino-small | 782 | 8 | 2 |
| 3964 | OVA_Lung | 1545 | 10935 | 2 | 3844 | kdd_internet_usage | 10108 | 68 | 2 |
| 4000 | OVA_Ovary | 1545 | 10935 | 2 | 359991 | kick | 72983 | 32 | 2 |
| 3980 | OVA_Prostate | 1545 | 10936 | 2 | 3672 | kin8nm | 8192 | 8 | 2 |
| 359971 | PhishingWebsites | 11055 | 30 | 2 | 190392 | madeline | 3140 | 259 | 2 |
| 361342 | Run_or_walk_informat | 88588 | 6 | 2 | 9976 | madelon | 2600 | 500 | 2 |
| 359975 | Satellite | 5100 | 36 | 2 | 3483 | mammography | 11183 | 6 | 2 |
| 125968 | SpeedDating | 8378 | 120 | 2 | 3907 | mc1 | 9466 | 38 | 2 |
| 361339 | Titanic | 2201 | 3 | 2 | 3623 | meta | 528 | 21 | 2 |
| 190411 | ada | 4147 | 48 | 2 | 3899 | mozilla4 | 15545 | 5 | 2 |
| 359983 | adult | 48842 | 14 | 2 | 3749 | no2 | 500 | 7 | 2 |
| 3600 | ailerons | 13750 | 40 | 2 | 359980 | nomao | 34465 | 118 | 2 |
| 190412 | arcene | 100 | 10000 | 2 | 167120 | numerai28.6 | 96320 | 21 | 2 |
| 3812 | arsenic-female-bladd | 559 | 4 | 2 | 190137 | ozone-level-8hr | 2534 | 72 | 2 |
| 9909 | autoUniv-au1-1000 | 1000 | 20 | 2 | 361341 | parity5_plus_5 | 1124 | 10 | 2 |
| 359982 | bank-marketing | 45211 | 16 | 2 | 3667 | pbcseq | 1945 | 18 | 2 |
| 3698 | bank32nh | 8192 | 32 | 2 | 3918 | pc1 | 1109 | 21 | 2 |
| 3591 | bank8FM | 8192 | 8 | 2 | 3919 | pc2 | 5589 | 36 | 2 |
| 359955 | blood-transfusion-se | 748 | 4 | 2 | 3903 | pc3 | 1563 | 37 | 2 |
| 3690 | boston_corrected | 506 | 20 | 2 | 359958 | pc4 | 1458 | 37 | 2 |
| 359968 | churn | 5000 | 20 | 2 | 190410 | philippine | 5832 | 308 | 2 |
| 146819 | climate-model-simula | 540 | 20 | 2 | 168350 | phoneme | 5404 | 5 | 2 |
| 3793 | colleges_usnews | 1302 | 33 | 2 | 3616 | pm10 | 500 | 7 | 2 |
| 3627 | cpu_act | 8192 | 21 | 2 | 3735 | pollen | 3848 | 5 | 2 |
| 3601 | cpu_small | 8192 | 12 | 2 | 3618 | puma32H | 8192 | 32 | 2 |
| 168757 | credit-g | 1000 | 20 | 2 | 3681 | puma8NH | 8192 | 8 | 2 |
| 14954 | cylinder-bands | 540 | 39 | 2 | 359956 | qsar-biodeg | 1055 | 41 | 2 |
| 3668 | delta_ailerons | 7129 | 5 | 2 | 9959 | ringnorm | 7400 | 20 | 2 |
| 3684 | delta_elevators | 9517 | 6 | 2 | 3583 | rmftsa_ladata | 508 | 10 | 2 |
| 37 | diabetes | 768 | 8 | 2 | 43 | spambase | 4601 | 57 | 2 |
| 125920 | dresses-sales | 500 | 12 | 2 | 359972 | sylvine | 5124 | 20 | 2 |
| 9983 | eeg-eye-state | 14980 | 14 | 2 | 361340 | tokyo1 | 959 | 44 | 2 |
| 219 | electricity | 45312 | 8 | 2 | 9943 | twonorm | 7400 | 20 | 2 |
| 3664 | fri_c0_1000_5 | 1000 | 5 | 2 | 3786 | visualizing_soil | 8641 | 4 | 2 |
| 3747 | fri_c0_500_5 | 500 | 5 | 2 | 146820 | wilt | 4839 | 5 | 2 |
| 3702 | fri_c1_1000_50 | 1000 | 50 | 2 | 3712 | wind | 6574 | 14 | 2 |
| 3766 | fri_c2_1000_25 | 1000 | 25 | 2 | | | | | |

Table 5: Multi-class classification datasets used in TabRepo.

| Task ID | name | n | f | C | Task ID | name | n | f | C |
|---|---|---|---|---|---|---|---|---|---|
| 211986 | Diabetes130US | 101766 | 49 | 3 | 6 | letter | 20000 | 16 | 26 |
| 359970 | GesturePhaseSegmenta | 9873 | 32 | 5 | 359961 | mfeat-factors | 2000 | 216 | 10 |
| 360859 | Indian_pines | 9144 | 220 | 8 | 359953 | micro-mass | 571 | 1300 | 20 |
| 125921 | LED-display-domain-7 | 500 | 7 | 10 | 189773 | microaggregation2 | 20000 | 20 | 5 |
| 146800 | MiceProtein | 1080 | 81 | 8 | 359993 | okcupid-stem | 50789 | 19 | 3 |
| 361330 | Traffic_violations | 70340 | 20 | 3 | 28 | optdigits | 5620 | 64 | 10 |
| 168300 | UMIST_Faces_Cropped | 575 | 10304 | 20 | 30 | page-blocks | 5473 | 10 | 5 |
| 3549 | analcatdata_authorsh | 841 | 70 | 4 | 32 | pendigits | 10992 | 16 | 10 |
| 3560 | analcatdata_dmft | 797 | 4 | 6 | 359986 | robert | 10000 | 7200 | 10 |
| 14963 | artificial-character | 10218 | 7 | 10 | 2074 | satimage | 6430 | 36 | 6 |
| 9904 | autoUniv-au6-750 | 750 | 40 | 8 | 359963 | segment | 2310 | 19 | 7 |
| 9906 | autoUniv-au7-1100 | 1100 | 12 | 5 | 9964 | semeion | 1593 | 256 | 10 |
| 9905 | autoUniv-au7-700 | 700 | 12 | 3 | 359987 | shuttle | 58000 | 9 | 7 |
| 11 | balance-scale | 625 | 4 | 3 | 41 | soybean | 683 | 35 | 19 |
| 2077 | baseball | 1340 | 16 | 3 | 45 | splice | 3190 | 60 | 3 |
| 359960 | car | 1728 | 6 | 4 | 168784 | steel-plates-fault | 1941 | 27 | 7 |
| 9979 | cardiotocography | 2126 | 35 | 10 | 3512 | synthetic_control | 600 | 60 | 6 |
| 359959 | cmc | 1473 | 9 | 3 | 125922 | texture | 5500 | 40 | 11 |
| 359957 | cnae-9 | 1080 | 856 | 9 | 190146 | vehicle | 846 | 18 | 4 |
| 146802 | collins | 1000 | 23 | 30 | 9924 | volcanoes-a2 | 1623 | 3 | 5 |
| 359977 | connect-4 | 67557 | 42 | 3 | 9925 | volcanoes-a3 | 1521 | 3 | 5 |
| 168909 | dilbert | 10000 | 2000 | 5 | 9926 | volcanoes-a4 | 1515 | 3 | 5 |
| 359964 | dna | 3186 | 180 | 3 | 9927 | volcanoes-b1 | 10176 | 3 | 5 |
| 359954 | eucalyptus | 736 | 19 | 5 | 9928 | volcanoes-b2 | 10668 | 3 | 5 |
| 3897 | eye_movements | 10936 | 27 | 3 | 9931 | volcanoes-b5 | 9989 | 3 | 5 |
| 168910 | fabert | 8237 | 800 | 7 | 9932 | volcanoes-b6 | 10130 | 3 | 5 |
| 359969 | first-order-theorem- | 6118 | 51 | 6 | 9920 | volcanoes-d1 | 8753 | 3 | 5 |
| 14970 | har | 10299 | 561 | 6 | 9923 | volcanoes-d4 | 8654 | 3 | 5 |
| 3481 | isolet | 7797 | 617 | 26 | 9915 | volcanoes-e1 | 1183 | 3 | 5 |
| 211979 | jannis | 83733 | 54 | 4 | 9960 | wall-robot-navigatio | 5456 | 24 | 4 |
| 359981 | jungle_chess_2pcs_ra | 44819 | 6 | 3 | 58 | waveform-5000 | 5000 | 40 | 3 |
| 9972 | kr-vs-k | 28056 | 6 | 18 | 361345 | wine-quality-red | 1599 | 11 | 6 |
| 2076 | kropt | 28056 | 6 | 18 | 359974 | wine-quality-white | 4898 | 11 | 7 |
| 361344 | led24 | 3200 | 24 | 10 | | | | | |

Table 6: Regression datasets used in TabRepo.

| Task ID | name | n | f | Task ID | name | n | f |
|---|---|---|---|---|---|---|---|
| 233212 | Allstate_Claims_Seve | 188318 | 130 | 359936 | elevators | 16599 | 18 |
| 359938 | Brazilian_houses | 10692 | 12 | 359952 | house_16H | 22784 | 16 |
| 233213 | Buzzinsocialmedia_Tw | 583250 | 77 | 359951 | house_prices_nominal | 1460 | 79 |
| 360945 | MIP-2016-regression | 1090 | 144 | 359949 | house_sales | 21613 | 21 |
| 233215 | Mercedes_Benz_Greene | 4209 | 376 | 359946 | pol | 15000 | 48 |
| 167210 | Moneyball | 1232 | 14 | 359930 | quake | 2178 | 3 |
| 359941 | OnlineNewsPopularity | 39644 | 59 | 359931 | sensory | 576 | 11 |
| 359948 | SAT11-HAND-runtime-r | 4440 | 116 | 359932 | socmob | 1156 | 5 |
| 317614 | Yolanda | 400000 | 100 | 359933 | space_ga | 3107 | 6 |
| 359944 | abalone | 4177 | 8 | 359934 | tecator | 240 | 124 |
| 359937 | black_friday | 166821 | 9 | 359939 | topo_2_1 | 8885 | 266 |
| 359950 | boston | 506 | 13 | 359945 | us_crime | 1994 | 126 |
| 359942 | colleges | 7063 | 44 | 359935 | wine_quality | 6497 | 11 |
| 233211 | diamonds | 53940 | 9 | 359940 | yprop_4_1 | 8885 | 251 |

Table 7: Model versions.

| model | benchmarked | latest | package |
|---|---|---|---|
| LightGBM | 3.3.5 | 4.0.0 | lightgbm |
| XGBoost | 1.7.6 | 2.0.0 | xgboost |
| CatBoost | 1.2.1 | 1.2.2 | catboost |
| RandomForest | 1.1.1 | 1.3.1 | scikit-learn |
| ExtraTrees | 1.1.1 | 1.3.1 | scikit-learn |
| MLP | 2.0.1 | 2.0.1 | torch |

### D.3 MLP

```
{
    'learning_rate': Real(1e-4, 3e-2, default=3e-4, log=True),
    'weight_decay': Real(1e-12, 0.1, default=1e-6, log=True),
    'dropout_prob': Real(0.0, 0.4, default=0.1),
    'use_batchnorm': Categorical(False, True),
    'num_layers': Int(1, 5, default=2),
    'hidden_size': Int(8, 256, default=128),
    'activation': Categorical('relu', 'elu'),
}
```

### D.4 CATBOOST

```
{
    'learning_rate': Real(lower=5e-3, upper=0.1, default=0.05, log=True),
    'depth': Int(lower=4, upper=8, default=6),
    'l2_leaf_reg': Real(lower=1, upper=5, default=3),
    'max_ctr_complexity': Int(lower=1, upper=5, default=4),
    'one_hot_max_size': Categorical(2, 3, 5, 10),
    'grow_policy': Categorical("SymmetricTree", "Depthwise")
}
```

### D.5 LIGHTGBM

```
{
    'learning_rate': Real(lower=5e-3, upper=0.1, default=0.05, log=True),
    'feature_fraction': Real(lower=0.4, upper=1.0, default=1.0),
    'min_data_in_leaf': Int(lower=2, upper=60, default=20),
    'num_leaves': Int(lower=16, upper=255, default=31),
    'extra_trees': Categorical(False, True),
}
```

### D.6 XGBOOST

```
{
    'learning_rate': Real(lower=5e-3, upper=0.1, default=0.1, log=True),
    'max_depth': Int(lower=4, upper=10, default=6),
    'min_child_weight': Real(0.5, 1.5, default=1.0),
    'colsample_bytree': Real(0.5, 1.0, default=1.0),
    'enable_categorical': Categorical(True, False),
}
```

### D.7 EXTRA-TREES

For all Extra Trees models we use 300 trees.

```
{
    'max_leaf_nodes': Int(5000, 50000),
    'min_samples_leaf': Categorical(1, 2, 3, 4, 5, 10, 20, 40, 80),
    'max_features': Categorical('sqrt', 'log2', 0.5, 0.75, 1.0)
}
```

### D.8 RANDOM-FOREST

For all Random Forest models we use 300 trees.

```
{
    'max_leaf_nodes': Int(5000, 50000),
    'min_samples_leaf': Categorical(1, 2, 3, 4, 5, 10, 20, 40, 80),
    'max_features': Categorical('sqrt', 'log2', 0.5, 0.75, 1.0)
}
```

Table 8: AutoML framework versions.

| framework | benchmarked | latest | package |
|---|---|---|---|
| AutoGluon | 0.8.2 | 0.8.2 | autogluon |
| auto-sklearn | 0.15.0 | 0.15.0 | auto-sklearn |
| auto-sklearn 2 | 0.15.0 | 0.15.0 | auto-sklearn |
| FLAML | 1.2.4 | 2.1.0 | flaml |
| H2O AutoML | 3.40.0.4 | 3.42.0.3 | h2o |
| LightAutoML | 0.3.7.3 | 0.3.7.3 | lightautoml |

## E  AUTOML FRAMEWORK DETAILS

For each AutoML framework we attempted to use the latest available versions where possible. The precise versions used for each framework are documented in Tab. 8. For FLAML, version 2.0 released after we had ran the experiments.

### E.1  AUTOML FRAMEWORK FULL RESULTS

Refer to the supplementary file results_ranked_automl.csv for the complete set of results for all AutoML systems on all tasks. This file includes the training runtime, inference time (per row), and metric error for each task.

### E.2  AUTO-SKLEARN 2

**Meta-Learning.**  Auto-Sklearn 2 uses meta-learning to improve the quality of its results. Since the datasets used to train its meta-learning algorithm are present in TabRepo, the performance of Auto-Sklearn 2 may be overly optimistic as it may be choosing to train model hyperparameters known to achieve strong test scores on a given dataset. This issue is detailed in section 5.3.3 of Gijsbers et al. (2022). Following Gijsbers et al. (2022), we ultimately decide to keep Auto-Sklearn 2's results as a useful comparison point.

**Regression.**  Auto-Sklearn 2 is incompatible with regression tasks. For regression tasks, we use the result from Auto-Sklearn 1.

## F  ADDITIONAL RESULTS WITH MORE MODELS

Here, we report results for additional models. In particular, we consider:

- A linear model
- A K-nearest neighbor model (Cover & Hart, 1967)
- TabPFN model (Hollmann et al., 2022) which is transformer model for tabular data pretrained on a collection of artificial datasets that performs attention over rows
- FT-Transformer (Gorishniy et al., 2021) which is a transformer trained on a dataset at hand and performs attention over columns

For TabPFN and FT-Transformer, we measure results on a g4.2xlarge instance. We run only the default configuration for FT-transformer due to the large training cost to obtain results on all tasks on a GPU machine, we also ran a single configuration for TabPFN.

We report those results separately because 1) as opposed to the previous collection of models, some models in this collection fail and requires imputation 2) some models requires an additional GPU as opposed to the models presented in the main sections which pose different hardware constraint cost.

Some of the models fails because of algorithm errors (for instance TabPFN only supports 100 features currently) or hardware errors (out-of-memory errors in case of KNN for instance). In case of failure, we impute the model predictions with the baseline used when portfolio configuration times out (e.g. the default configuration of Extra-trees), this baseline always take less than 5 seconds to run.

Table 9: Results with additional models defined in Section F. Normalized-error, rank, training and inference time are averaged over all tasks given 4h training budget.

| method | normalized-error | rank | time fit (s) | time infer (s) |
|---|---|---|---|---|
| Portfolio (ensemble) | 0.362 | 174.6 | 6597.5 | 0.061 |
| AutoGluon best | 0.389 | 208.2 | 5583.1 | 0.062 |
| Portfolio | 0.437 | 236.6 | 6597.5 | 0.013 |
| Autosklearn2 | 0.455 | 243.5 | 14415.9 | 0.013 |
| AutoGluon high | 0.463 | 283.3 | 5460.8 | 0.002 |
| Lightautoml | 0.466 | 246.1 | 9173.9 | 0.298 |
| Flaml | 0.513 | 317.8 | 14267.0 | 0.002 |
| CatBoost (tuned + ensemble) | 0.524 | 267.3 | 9065.2 | 0.012 |
| H2oautoml | 0.526 | 337.0 | 13920.1 | 0.002 |
| AutoGluon medium | 0.527 | 311.0 | 371.8 | 0.001 |
| CatBoost (tuned) | 0.534 | 284.7 | 9065.2 | 0.002 |
| LightGBM (tuned + ensemble) | 0.534 | 268.7 | 3528.9 | 0.010 |
| LightGBM (tuned) | 0.566 | 304.2 | 3528.9 | 0.001 |
| CatBoost (default) | 0.586 | 341.2 | 456.8 | 0.002 |
| MLP (tuned + ensemble) | 0.594 | 402.5 | 5771.8 | 0.098 |
| XGBoost (tuned + ensemble) | 0.628 | 357.9 | 4972.7 | 0.013 |
| MLP (tuned) | 0.634 | 451.9 | 5771.8 | 0.014 |
| XGBoost (tuned) | 0.638 | 376.5 | 4972.7 | 0.002 |
| FTTransformer (default) | 0.690 | 532.1 | 567.4 | 0.003 |
| LightGBM (default) | 0.714 | 491.5 | 55.7 | 0.001 |
| XGBoost (default) | 0.734 | 522.2 | 75.1 | 0.002 |
| MLP (default) | 0.772 | 629.4 | 38.2 | 0.015 |
| ExtraTrees (tuned + ensemble) | 0.782 | 544.2 | 538.3 | 0.001 |
| ExtraTrees (tuned) | 0.802 | 572.5 | 538.3 | 0.000 |
| RandomForest (tuned + ensemble) | 0.803 | 578.3 | 1512.2 | 0.001 |
| RandomForest (tuned) | 0.816 | 598.0 | 1512.2 | 0.000 |
| TabPFN (default) | 0.837 | 731.9 | 3.8 | 0.016 |
| LinearModel (tuned + ensemble) | 0.855 | 873.8 | 612.4 | 0.038 |
| LinearModel (tuned) | 0.862 | 891.6 | 612.4 | 0.006 |
| ExtraTrees (default) | 0.883 | 788.6 | 3.0 | 0.000 |
| RandomForest (default) | 0.887 | 773.9 | 13.8 | 0.000 |
| LinearModel (default) | 0.899 | 940.1 | 7.1 | 0.014 |
| KNeighbors (tuned + ensemble) | 0.928 | 980.8 | 12.0 | 0.001 |
| KNeighbors (tuned) | 0.937 | 1016.5 | 12.0 | 0.000 |
| KNeighbors (default) | 0.973 | 1149.1 | 0.6 | 0.000 |

Table 10: Win rate comparison with additional models defined in Section F for 4 hour time limit.

| method | winrate | > | < | = | time fit (s) | time infer (s) | loss (rescaled) | rank |
|---|---|---|---|---|---|---|---|---|
| Portfolio (ensemble) | 0.500 | 0 | 0 | 200 | 6597.5 | 0.061 | 0.239 | 3.115 |
| AutoGluon best | 0.425 | 80 | 110 | 10 | 5583.1 | 0.062 | 0.290 | 3.442 |
| Autosklearn2 | 0.350 | 68 | 128 | 4 | 14415.9 | 0.013 | 0.404 | 4.360 |
| Lightautoml | 0.287 | 56 | 141 | 3 | 9173.9 | 0.298 | 0.434 | 4.625 |
| CatBoost (tuned + ensemble) | 0.245 | 47 | 149 | 4 | 9065.2 | 0.012 | 0.506 | 5.008 |
| Autosklearn | 0.290 | 56 | 140 | 4 | 14413.6 | 0.009 | 0.515 | 5.045 |
| Flaml | 0.295 | 56 | 138 | 6 | 14267.0 | 0.002 | 0.533 | 5.090 |
| H2oautoml | 0.223 | 41 | 152 | 7 | 13920.1 | 0.002 | 0.565 | 5.315 |

As one can see in Tab. 9, FT-Transformer performs in-between MLPs and the best boosted trees methods. Regarding TabPFN, the method does not reach the performance of top methods yet which is due to high failure rates due to current method limitations on large datasets[5] and also due to the method not being able to currently exploit well large number of rows.

The results of portfolio improves given the additional model diversity which can be seen by looking at Tab. 10 which reports the win-rate against AutoML baselines. In particular, the win rate is improved from 53.5% to 57.5%.

---

[5]The failure rate is $\approx 30\%$ as the method only supports 100 features and 10 classes.

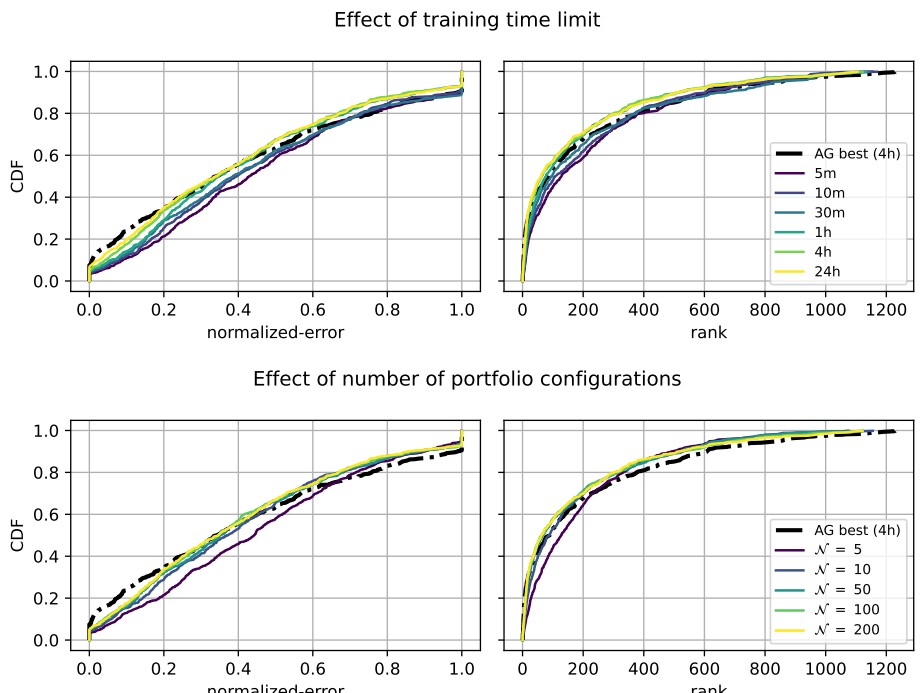

Figure 5: Effect on performance of fitting time (top) and number of portfolio configurations (bottom). In the second case, portfolios are fitted with a 4h fitting budget.

# G  ADDITIONAL RESULTS PORTFOLIOS

## G.1  PORTFOLIO WIN-RATE COMPARISON

We calculate win-rates, re-scaled loss, and average ranks between the Portfolio and the AutoML systems in Tab. 12 and Tab. 2 for 1 and 4 hour time limits respectively with the same evaluation protocol as Erickson et al. (2020). In both cases, Portfolio achieves the best win-rate, re-scaled loss, and average rank across all methods at the given time constraint.

## G.2  EFFECT OF RUNTIME AND NUMBER OF PORTFOLIO CONFIGURATIONS

In Fig. 5, we show the effect of increasing the time budget bound and the number of portfolio configuration bound $\mathcal{N}$. Increasing the fitting time limit yields constant improvement however increasing the number of portfolio configurations provides quickly diminishing returns given that a large number of configurations can rarely be evaluated given a reasonable time limit except for very small datasets.

## G.3  PERFORMANCE ON LOWER FITTING BUDGETS

In section 5, we reported results for 1h, 4h fitting budgets which are standard settings (Erickson et al., 2020; Gijsbers et al., 2022). Given space constraint, we only showed the full table for 4h results in the main, the results for 1h results is shown in Tab. 11. Here the anytime portfolio strategy matches AutoGluon on normalized-error and outperforms it on rank while having around 30% lower latency.

In Fig. 7, we also report the performance for 5m, 10m, 30m budgets in addition to 1h, 4h and 24h for both portfolio ensembles and AutoGluon. For budgets lower than 1h, we select portfolio configurations only among configurations whose runtime are under the constraint at least 95% of the time on available training datasets. For budgets lower than 1h, the performance of portfolio drops significantly. However, this result is overly pessimistic. Indeed, whenever the first configuration selected by the portfolio does not finish before the constraint, we return the result of a cheap baseline

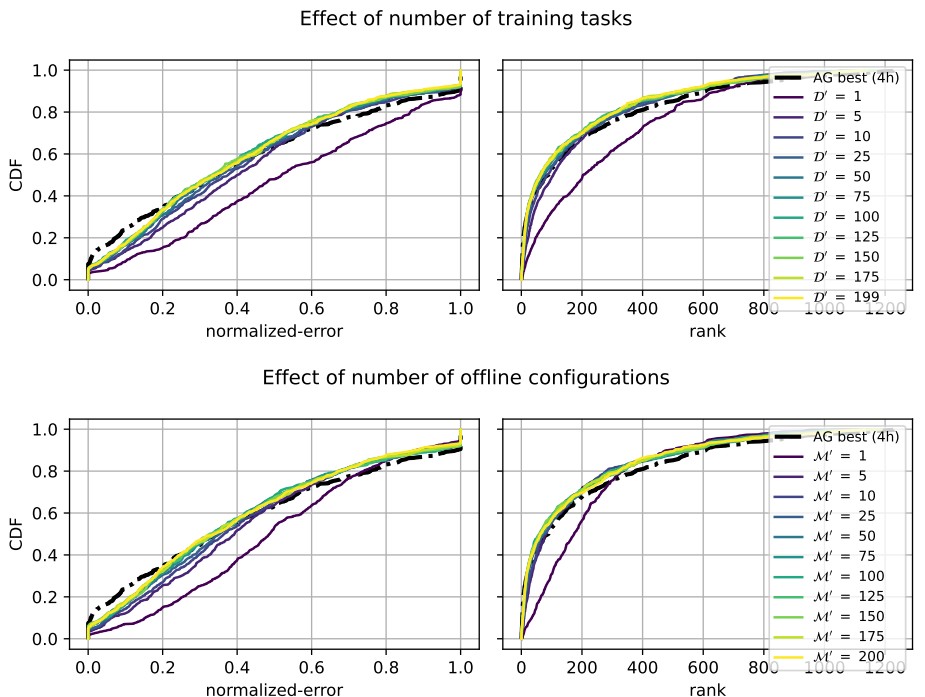

Figure 6: From top to bottom, effect of the number of training datasets and offline configurations per model family on distribution performance. All methods are fitted under a 4h fitting budget.

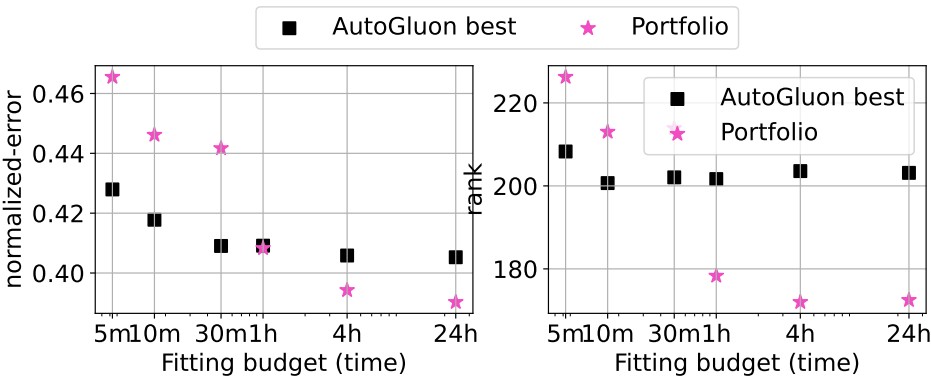

Figure 7: Scatter plot of average normalized error (left) and rank (right) against fitting smaller training time budget for portfolio and AutoGluon.

Table 11: Normalized-error, rank, training and inference time averaged over all tasks given 1h training budget.

| method | normalized-error | rank | time fit (s) | time infer (s) |
|---|---|---|---|---|
| Portfolio (ensemble) | 0.408 | 178.3 | 2435.0 | 0.023 |
| AutoGluon best | 0.409 | 201.6 | 2283.6 | 0.033 |
| AutoGluon high | 0.477 | 273.1 | 2201.2 | 0.002 |
| Autosklearn2 | 0.494 | 254.1 | 3611.2 | 0.010 |
| Lightautoml | 0.503 | 246.4 | 3007.9 | 0.099 |
| H2oautoml | 0.547 | 316.5 | 3572.8 | 0.002 |
| Flaml | 0.550 | 330.1 | 3622.9 | 0.001 |
| AutoGluon medium | 0.551 | 306.5 | 270.5 | 0.001 |
| LightGBM (tuned + ensemble) | 0.563 | 260.5 | 1622.5 | 0.009 |
| CatBoost (tuned + ensemble) | 0.576 | 281.8 | 2873.5 | 0.005 |
| CatBoost (tuned) | 0.588 | 296.0 | 2884.7 | 0.002 |
| LightGBM (tuned) | 0.597 | 300.7 | 1627.9 | 0.002 |
| CatBoost (default) | 0.614 | 332.4 | 443.7 | 0.002 |
| MLP (tuned + ensemble) | 0.622 | 405.1 | 2560.2 | 0.107 |
| MLP (tuned) | 0.653 | 447.2 | 2559.8 | 0.014 |
| XGBoost (tuned + ensemble) | 0.662 | 356.5 | 1860.1 | 0.012 |
| XGBoost (tuned) | 0.675 | 379.1 | 1856.6 | 0.002 |
| LightGBM (default) | 0.747 | 478.7 | 54.2 | 0.001 |
| XGBoost (default) | 0.768 | 509.4 | 73.2 | 0.002 |
| MLP (default) | 0.782 | 611.3 | 39.7 | 0.015 |
| ExtraTrees (tuned + ensemble) | 0.799 | 525.4 | 386.9 | 0.001 |
| ExtraTrees (tuned) | 0.818 | 553.7 | 386.8 | 0.000 |
| RandomForest (tuned + ensemble) | 0.818 | 559.0 | 676.4 | 0.001 |
| RandomForest (tuned) | 0.830 | 575.6 | 671.8 | 0.000 |
| ExtraTrees (default) | 0.889 | 762.3 | 3.8 | 0.000 |
| RandomForest (default) | 0.896 | 749.4 | 17.5 | 0.000 |

Table 12: Win rate comparison for 1 hour time limit with the same approach used as for Tab. 2.

| method | winrate | > | < | = | time fit (s) | time infer (s) | loss (rescaled) | rank |
|---|---|---|---|---|---|---|---|---|
| Portfolio (ensemble) (1h) | **0.500** | | | 200 | 2434.9 | 0.023 | **0.250** | **3.257** |
| AutoGluon best (1h) | 0.487 | 95 | 100 | 5 | 2283.6 | 0.033 | 0.273 | 3.308 |
| Autosklearn2 (1h) | 0.380 | 74 | 122 | 4 | 3611.2 | 0.010 | 0.392 | 4.280 |
| Lightautoml (1h) | 0.307 | 59 | 136 | 5 | 3007.9 | 0.099 | 0.411 | 4.423 |
| CatBoost (tuned + ensemble) (1h) | 0.233 | 45 | 152 | 3 | 2876.2 | 0.004 | 0.518 | 5.122 |
| H2oautoml (1h) | 0.263 | 51 | 146 | 3 | 3572.8 | 0.002 | 0.513 | 5.128 |
| Autosklearn (1h) | 0.310 | 60 | 136 | 4 | 3612.0 | 0.007 | 0.545 | 5.185 |
| Flaml (1h) | 0.278 | 53 | 142 | 5 | 3622.9 | 0.001 | 0.559 | 5.298 |

as we do not store all model checkpoints whereas AutoGluon instead uses the best checkpoint found until that point. To allow simulation under those cheaper settings, one would have to store more checkpoints per model (for instance at a 10 minute frequency or at 5m, 10m, 30m and 1h time points as done in Borchert et al. (2022)) but we decided against this option as the storage cost of TabRepo is already significant (roughly 107 GB).

