# OpenReview forum: "TabRepo: A Large Scale Repository of Tabular Model Evaluations and its AutoML Applications"
_ICLR.cc/2024/Conference — Submitted to ICLR 2024_

### Official Review · Reviewer_EeeT · 2023-10-29

**Soundness:** 2 fair
**Presentation:** 3 good
**Contribution:** 2 fair
**Rating:** 3
**Confidence:** 4

**Summary:**

The paper proposes TabRepo -- a dataset of predictions and metrics for 1206 hyperparameter configurations of 6 models (i.e. 201 configuration per model) on 200 classification and regression tasks (to clarify, these 200 tasks are taken from existing public benchmarks).

The paper demonstrates that TabRepo can be useful for:
- analyzing whether hyperparameter tuning and ensembling can help traditional models outperform modern AutoML systems;
- analyzing ensembling strategies by using the published model predictions without retraining these models (*"at no cost"*);
- performing *"transfer learning"* (in this paper, this term describes the usage of results obtained on some tasks to inform the choice of models/hyperparameters/etc. on other tasks); as an example of this, the paper shows that portfolio learning outperforms existing AutoML approaches.

**Strengths:**

- The paper is easy to follow.
- The training, evaluation, ensembling and hyperparameter tuning protocols are clear and transparent.
- The evaluation of the "Portfolio learning" technique and its positive results is interesting.
- TabRepo is the largest dataset in its niche.
- Many modern AutoML algorithms are covered.

**Weaknesses:**

*(a quick comment on the selected confidence level: I am not a big expert in the whole landspace of AutoML papers, though I am familiar with this type of methods; as for all other aspects, I am fully familiar with them)*


**(A) Regarding datasets, in my opinion, the "quality vs. quantity" balance should be improved.** I believe that, for the field of tabular data, it is time to raise the bar in terms of dataset quality and to compose benchmarks that will have more chances to generalize to real world problems. After a quick review, I noticed the following datasets that can make the benchmark biased in ways that a hypothetical practitioner would not approve:
- volcanoes-{a2,a3,a4,b1,b2,b5,b6,d1,d4,e1} -- the real world is not 10x biased towards tabular datasets about volcanoes with 3 features and 5 classes, but the benchmark is biased in this way.
- wine-quality-{red,white} -- see the previous bullet.
- fri_c0_1000_5, fri_c0_500_5 and 8 more similar tasks (10 in total) -- also seems to be a set of closely related problems as in previous bullets.
- optdigits -- I think that computer vision problems should not be included in general tabular benchmarks, or should be presented in a separate group.
- kr-vs-k -- I think that deterministic game-based problems (here, chess) should not be included in general tabular benchmarks, or should be presented in a separate group.
- etc.

Perhaps, works like `[1]` can be a source of more realistic datasets (worth mentioning: `[1]` is a bit limited in terms of dataset sizes, so other works like `[2]` may also be worth considering).

**(B) I think that models should be more diverse.** The current set of models is strongly biased towards:
- tree-based models (all models except for MLP)
- ensemble-like models (all models except for MLP)

I am afraid that this may limit the potential of TabRepo in terms of what kind of analysis it allows conducting and what results it allows uncovering. I suggest considering the following:
- Adding one linear model.
- Adding one non-parametric model (e.g. kNN or modern kNN-like models), at least on datasets where it is possible.
- Adding one modern parametric DL model (note that competitive parametric DL models are not necessarily heavy transformers `[3]`).
- Adding one modern non-parametric DL model.
- Keeping no more than two gradient boostings (personally, I would prefer just one, again, to reduce tge bias, given that there is also RandomForest).
- Excluding ExtraTrees.

Also, I appreciate that there are various opinions on whether tabular DL models are worth attention. However, if DL models are not well presented, then the benchmark should be positioned as Classic-ML-only benchmark, but not as a general benchmark. Otherwise, some readers may have wrong expectations from the title and the abstract.

**(C) In my opinion, the paper may need bigger stories (bigger than Section 4 and Section 5) to support the proposed dataset.** My understanding is that it is (implicitly) suggested that TabRepo will help others to uncover and tell big/novel/non-trivial stories. However, compared to mainstream dataset-oriented works like `[1]` (where it is easy to imagine a wide target audience and a potential range of works based on the proposed benchmark), TabRepo seems to be more niche, and, to me, it is not immediately obvious how TabRepo can be used to obtain novel results. This is why, in this specific case, I expect the proposed dataset to be supported by at least one strong self-sufficient finding.

I would like to add that:
- I appreciate the stories told in Section 4 and Section 5, there is nothing wrong with them. However, they are not positioned as founding elements of the paper and, indeed, it may be too early to position them as such.
- If (A) and (B) are perfectly addressed, then (C) will not be a blocker, at least not for me.

**(D) Other things:**
- The story in the introduction may be a bit polarizing. I mean things like "their performance has saturated and state-of-the-art methods now leverage AutoML techniques", *"AutoML solutions currently dominate tabular prediction benchmarks"*, etc. I embraced the suggested perspective for the review, but overall, I don't share it, and I can imagine how this can trigger big discussions.
- Not a big issue, but personally, I find the "at no cost" wording a bit controversial, I would probably avoid it or somehow make it softer. The reported findings have non-zero cost, and the ability to use a public dataset at no cost is a usual property of public datasets. Perhaps, I am missing something here, but sharing this impression just in case.

**References**

- `[1]` "Why do tree-based models still outperform deep learning on tabular data?" Grinsztajn et al.
- `[2]` "TabR: Tabular Deep Learning Meets Nearest Neighbors in 2023" Gorishniy et al.
- `[3]` "On Embeddings for Numerical Features in Tabular Deep Learning" Gorishniy et al.

**Questions:**

See weaknesses.

---

> ### Author Response · Authors · 2023-11-20
> **Answer**
>
> Thank you very much for your constructive and thorough review.
>
> ```
> (A) Regarding datasets, in my opinion, the "quality vs. quantity" balance should be improved.
> ```
> We took the union of the collection from the AutoMLBenchmark (Gijsbers et al., 2022) and from the Auto-Sklearn 2 paper (Feurer et al., 2020), the exact rationale for dataset selection is currently detailed in Appendix C.
> Our motivation for extending those suites was that both works are prominent in the field of AutoML and in particular Gijsbers 2022 is the current most extensive empirical evaluation of AutoML methods.
> We agree that the study [1] is also interesting but we believe the current collection covers sufficient variety (it contains the majority of datasets from [1] for instance).
>
> ```
> (B) I think that models should be more diverse. The current set of models is strongly biased towards: tree-based models (all models except for MLP) ensemble-like models (all models except for MLP). I am afraid that this may limit the potential of TabRepo
> ```
>
> We agree that more methods are worth adding, in particular DL methods, to increase model diversity and the usefulness of TabRepo.
> Following your suggestion, we added: one linear model, one non-parametric model (KNN), one modern parametric DL model (FT-transformer), one modern non-parametric DL model (TabPFN).
> The results are described in the main response with a table. They follow the observations done in prior work [1,3], e.g. "modern" DL models are in between the performance of the best boosted trees methods and MLPs.
>
> We do not think having only one gradient boosting method would be better for the dataset given that we have 3 DL methods now and boosted trees are performing better on average (the performance of the portfolio degrades significantly when removing CatBoost or LightGBM). We believe the methods are quite balanced between DL and trees with those additions.
>
> Regarding excluding extra-trees, we also do not think it would make the dataset better. The method is useful in itself in particular given that it is extremely fast and can be useful in case one wants to find solution with very low fitting time (as can be seen in Fig 1).
>
> ```
> (C) My understanding is that it is (implicitly) suggested that TabRepo will help others to uncover and tell big/novel/non-trivial stories. However, compared to mainstream dataset-oriented works like [1] (where it is easy to imagine a wide target audience and a potential range of works based on the proposed benchmark), TabRepo seems to be more niche, and, to me, it is not immediately obvious how TabRepo can be used to obtain novel results.
> ```
> We believe we demonstrated at least one impactful use-case for TabRepo which is reaching or outperforming the state-of-the-art of tabular predictions by simply considering ensemble of portfolio configurations. We agree that [1] provides valuable insight on the performance of DL versus tree-based methods. However, it serves a different purpose: it does not compare with SOTA tabular prediction methods (such as AutoGluon or AutoSklearn2 which use ensembling between different model families) and the dataset cannot be used to reach performance close to SOTA without spending significant compute given that all model would have to be retrained to evaluate the performance of ensembles. We believe our paper is serving more than a niche given that reaching SOTA performance at low cost is very important for research.
>
> ```
> (D1) "The story in the introduction may be a bit polarizing. I mean things like "their performance has saturated and state-of-the-art methods now leverage AutoML techniques", "AutoML solutions currently dominate tabular prediction benchmarks", etc. I embraced the suggested perspective for the review, but overall, I don't share it."
> ```
> Thank you for pointing this in a constructive fashion. Our intent was not to be polarizing and we highly appreciate your feedback.
> We understand that the sentences may have been polarizing as it opposes base models with AutoML systems whereas both work together (AutoML systems become better with any base models improvement). We reformulated the sentences and removed the term "saturated", "dominate" which we see as indeed potentially polarizing, we hope the paragraph now reads better.
>
> ```
> (D2) "I find the "at no cost" wording a bit controversial".
> ```
> We agree with your point that "at no cost" is potentially misleading and we reworded the sentence to "at marginal cost". We believe that "at marginal cost" is suited given that running the experiments of the paper takes 2 hours of compute with TabRepo instead of 27K hours if models are retrained from scratch.

---

### Official Review · Reviewer_XQfD · 2023-11-01

**Soundness:** 3 good
**Presentation:** 3 good
**Contribution:** 3 good
**Rating:** 6
**Confidence:** 4

**Summary:**

The paper introduces a big dataset of tabular model predictions. It showcases a few use cases of such dataset like: offline evaluation of hyperparameter tuning, hyperparameter transfer (portfolio learning). It shows that a simple portfolio learning method using this dataset outperforms state-of-the-art AutoML system on a standard benchmark.

**Strengths:**

- The paper is very well written.
- The experimental setup is sound: proper baselines, standard and relevant benchmark.
- The idea of sharing a large set of model evaluations is interesting, potentially practical and extensible.
- Performing on par with SoTA AutoML systems with a simple model selection technique from a proposed dataset.
- The investigation into how much data is needed for efficient transfer is insightful.

**Weaknesses:**

- A set of models is rather small. One potentially interesting
  extension could be adding more mainstream DL techniques (besides
  transformers, which are considerably slower, as correctly noted in
  the limitations section). Extending MLPs with regularization
  techniques from `[1]` could make the results more "modern" in the DL
  part of tabular models. MLPs with embeddings for continuous features
  from `[2]` is another potential candidate for a more "modern" but
  still fast DL method.
- It is unclear how the results would transfer to a more
  out-of-distribution datasets. Would portfolio transfer work as well
  as AutoML or hyperparameter tuning on datasets that differ from the
  datasets present in the benchmark in some aspects. Seeing
  performance on disregarded larger datasets (discussed in appendix)
  could shed more light on practical applicability of TabRepo
- One limitation that should be discussed is the tradeoff between
  computational efficency and memory. TabRepo is a large dataset,
  this could introduce problems in practice and make AutoML systems preferable.

**References**:
- `[1]` Kadra, Arlind, et al. "Well-tuned simple nets excel on tabular datasets." Advances in neural information processing systems 34 (2021): 23928-23941.
- `[2]` Gorishniy, Yury, Ivan Rubachev, and Artem Babenko. "On embeddings for numerical features in tabular deep learning." Advances in Neural Information Processing Systems 35 (2022): 24991-25004.

**Questions:**

- Are the portfolios (selected models+hyperparameters) interpretable? (In a rough sense: there is a large MLP, small GBDT, heavily regularized MLP, etc.), or the portfolios are mostly random and change for different subsets (of datasets)?
- Could TabRepo results be used for a new, potentially OOD datasets (for example larger tabular datasets than present in the benchmark). How does zero shot portfolio transfer compares to AutoML and hyperparameter tuning on OOD datasets?

Minor remarks (mostly stylistic or notation):
- In the model bagging section there might be a slight misuse of the $[n] = \{1,...,n\}$ notation introduced earlier, where it is used as an index in $(X^{(\mathrm{train})}[b], y^{(\mathrm{train})}[b]), (X^{(\mathrm{val})}[b], y^{(\mathrm{val})}[b])$.
- In the same model bagging section and the next (Datasets, predictions and evaluation) you say that models are fitted by minimizing the losses, in case of binary classification it's AUC, is it directly optimized for all models, or is it just used as a metric (and the terms loss and metric are used interchangeably)?

---

> ### Author Response · Authors · 2023-11-20
> **Answer**
>
> We thank you for the review.
>
> ```
> One potentially interesting extension could be adding more mainstream DL techniques...
> ```
>
> We added two more "modern" DL models, FT-transformer from [2] and TabPFN, as it was also pointed out by Reviewer EeeT that this family was under-represented. The methods performs a bit better than MLPs and are a bit worse that tree-based methods on the studied datasets which is consistent with previous findings in the literature.
>
>
> ```
> It is unclear how the results would transfer to a more out-of-distribution datasets.
> ```
> Regarding the generalization to out-of-distribution datasets, we evaluate portfolios on one unseen dataset via leave-one-dataset-out cross-validation. Of-course, the unseen dataset is still drawn from the same "distribution" of OpenML datasets however this gives a diverse distribution (as can be seen in Table 4-5-6 in the appendix).
>
> ```
> Could TabRepo results be used for a new, potentially OOD datasets (for example larger tabular datasets than present in the benchmark). How does zero shot portfolio transfer compares to AutoML and hyperparameter tuning on OOD datasets?
> ```
> Yes, we initially ran our evaluations with all datasets including the larger ones but observed the same results. Consequently, we decided to filter them as one obtained the same conclusion at lower compute cost. With larger datasets, we observed the same behavior e.g. that portfolio were outperforming AutoML systems which were outperforming ensembles of models from single families.
>
> That being said, we believe one could improve over simple portfolio approaches which are agnostic to the dataset at hand by engineering dataset features. We chose to not include this approach in order to propose a simple baseline as we thought having a simple method perform on-par or better than current SOTA systems is a more interesting finding.
>
> ```
> One limitation that should be discussed is the tradeoff between computational efficency and memory. TabRepo is a large dataset, this could introduce problems in practice and make AutoML systems preferable.
> ```
> Thank for pointing this, we agree the tradeoff between efficiency and memory is important and we added a discussion of this point in the paper.
> As pointed out in the appendix TabRepo takes 107GB as such having to store it in memory would impose expensive hardware. We bypassed this issue by using a memmap datastructure which allows to load the model predictions on the fly from disk to memory and allows to run the experiments on commodity hardware (only ~20GB of memory are needed). We added a mention of this point in appendix A).
>
> Note that loading TabRepo is only required to perform portfolio selection, once those are learned an AutoML system can just store the list of models and re-use them on unseen tasks.
>
> ```
> Are the portfolios (selected models+hyperparameters) interpretable? (In a rough sense: there is a large MLP, small GBDT, heavily regularized MLP, etc.), or the portfolios are mostly random and change for different subsets (of datasets)?
> ```
> Yes indeed, the portfolios are interpretable and consistent when computed for different hold-out test dataset.
> This consistency is expected as each portfolio shares 198 datasets in common with one-another (due to leave-one-dataset-out).
>
> Every portfolio contains the exact same first two models from CatBoost and MLP families. The third model is always from the LightGBM model family although it is not always exactly the same configuration. This portfolio pick order is somehow expected as tree models and neural networks ensemble well due to their lower correlation (which can be seen in Fig 1) and thus a MLP configuration is picked despite being a weaker model individually than other options. The early models all have low learning rates and are above average in model size (ex: number of leaves, depth, layers), but otherwise have hyperparameters that avoid the extremes of the search spaces.
>
> Interestingly, we observe that all 6 model families are picked at least once within the first 12 models in the portfolios which indicates that the model diversity is beneficial.

---

### Official Review · Reviewer_PZnZ · 2023-11-01

**Soundness:** 3 good
**Presentation:** 3 good
**Contribution:** 1 poor
**Rating:** 3
**Confidence:** 3

**Summary:**

The paper provides a collection of predictions from a wide range of models over an extensive benchmark of 200 regression and classification datasets. The paper draws some conclusions about model performance in different families, and how the predictions can be used for post-hoc ensembling.

**Strengths:**

I agree with the paper that extensive benchmarking is quite expensive, and the AutoML benchmark in particular is expensive to run.
The paper is quite clearly written, and easy to follow.

**Weaknesses:**

It's unclear to me how the problem of expensive benchmarks is solved by the proposed repository; at best it can be a benchmark for ensemble strategies. Something similar has been done in "CMA-ES for Post Hoc Ensembling in AutoML: A Great Success and Salvageable Failure" by Purucker, though with much more involved ensembling methods.

This paper only uses a simple greedy method, similar to what is used in Autosklearn, or "Mining Robust Default Configurations for Resource-constrained AutoML" (Flaml zero shot) or "Learning Multiple Defaults for Machine Learning Algorithms" or "Learning hyperparameter optimization initializations".

Futhermore, the distinction and benefit over OpenML is not entirely clear. Figure 1, for example, could have been generated with the runs on OpenML, which contains 10M runs, compared to 200k runs in this paper (with the disclaimer that the runs are not a cross-product of models and datasets, i.e. not all models are evaluated on all datasets, though there is several "studies" that do exactly that).

The main reason that OpenML does not store predictions or probabilities is that this would be very storage intensive and there is no funding for it. Most works that do portfolio building have computed all of these metrics, though they are usually not shared since the storage overhead seems daunting.

**Questions:**

How large is TabRepo in GB?
Do you intent for TabRepo to have new models dynamically added, or do you want to fix thecurrent models?
What future uses do you see for TabRepo?
How does your work compare to "CMA-ES for Post Hoc Ensembling in AutoML"?

---

> ### Author Response · Authors · 2023-11-20
> **Answer**
>
> We thank you for your review.
>
> ```
> * It's unclear to me how the problem of expensive benchmarks is solved by the proposed repository; at best it can be a benchmark for ensemble strategies. Something similar has been done in "CMA-ES for Post Hoc Ensembling in AutoML: A Great Success and Salvageable Failure" by Purucker, though with much more involved ensembling methods.
> * This paper only uses a simple greedy method, similar to what is used in Autosklearn, or "Mining Robust Default Configurations for Resource-constrained AutoML" (Flaml zero shot) or "Learning Multiple Defaults for Machine Learning Algorithms" or "Learning hyperparameter optimization initializations".
> ```
> We agree that the ensembling method proposed in Purucker is very interesting. However, we chose to report only the performance of simple known methods (Caruana ensemble on top of greedy portfolio) because the the main point of our paper is to introduce a dataset and not a new method. In particular, we show that those simple known methods outperform or matche the performance of all AutoML systems and methods which we believe illustrate some of the benefit of the proposed dataset. That being said, we believe the work is very relevant to provide further improvement and added a reference in our paper.
>
> ```
> * The main reason that OpenML does not store predictions or probabilities is that this would be very storage intensive and there is no funding for it. Most works that do portfolio building have computed all of these metrics, though they are usually not shared since the storage overhead seems daunting.
> * Furthermore, the distinction and benefit over OpenML is not entirely clear. Figure 1, could have been generated with the runs on OpenML ...
>
> ```
> The storage is ~107GB (mentioned in the appendix C and F3) which is tractable. As you pointed out, most work does not store predictions and does not allow to efficiently measure the performance of ensemble configurations, this is a key contribution of our paper and dataset.
>
> Regarding the difference with OpenML, we agree that the first two figures could have been generated with current results of OpenML. However, the key difference of TabRepo is to also expose the model predictions. This is a critical aspect as it allows researchers to consider ensembles and consequently to simulate solutions that outperforms SOTA AutoML systems at little cost (just recomputing metrics after having retrieved existing predictions).
>
> We believe that the key findings of the paper that the performance of SOTA tabular methods can be matched with simple techniques (as you said using only simple greedy methods) is a valuable finding for the community. We also think that providing a way to reach SOTA at low compute cost can be highly valuable for future research.
>
> ```
> How large is TabRepo in GB? Do you intent for TabRepo to have new models dynamically added, or do you want to fix the current models? What future uses do you see for TabRepo? How does your work compare to "CMA-ES for Post Hoc Ensembling in AutoML"?
> ```
>
> Regarding your last question, we plan to add landmark models over time. As future use, we believe any work considering ensembling can leverage TabRepo, in particular we believe work considering multi-fidelity (e.g. increase the number of folds as the fidelity) or cross hyperparameter/model tuning (CASH) are both interesting research directions.

---

> > ### Comment · Reviewer_PZnZ · 2023-11-22
> > **Re Purucker**
> >
> > My point in bringing up Purucker was mostly that the kind of research that you want to enable with TabRepo is already being done, and that someone interested in this research could simply reach out to Purucker and ask for the dataset they used.
> > Similar is likely true for the other methods I mentioned. The reason that Purucker did not make their data immediately available is likely not because they want to keep it private, but because they didn't consider it a valuable research contribution.
> > The fact that none of the previous publications didn't bother to make the intermediate data available doesn't seem a strong enough reason to publish this one.

---

> > > ### Author Response · Authors · 2023-11-22
> > > **Response to Reviewer PZnZ**
> > >
> > > Thank you for your response.
> > >
> > > We would like to highlight several points.
> > >
> > > 1. The CMA-ES paper only evaluates the default models of AutoGluon and does not define search spaces, leading to only a limited number of configurations being considered.
> > > 2. It does not have the same models trained on every dataset as mentioned in the paper in section 3.1: "In the end, AutoGluon
> > > produced between 2 and 24 base models".
> > > 3. It does not contain regression datasets.
> > > 5. It does not provide equal time budget to all configs, instead giving 4 hours total for each task, with each config potentially using all of the available time, greatly impacting the strength of the models based on the order they are trained (or if they were trained at all). In our work, we ensure a 1 hour time budget for every config on every task, to eliminate these kinds of issues.
> > > 4. CMA-ES is concurrent work only published very recently.
> > > 5. CMA-ES contains (up to) 24 configs trained on 73 datasets, our work contains 1206 configs trained on 200 datasets (>100x more results).
> > >
> > > Re-using the CMA-ES prediction dataset for our work would be insufficient as it does not contain the same configs for each dataset required to create a zeroshot-HPO portfolio for unseen tasks, and it contains far fewer overall configurations that would lead to meaningfully worse results. Referring to Figure 4 in our work, this is equivalent to ~4 configurations per family, which would not be sufficient to outperform AutoGluon.
> > >
> > > While prior work may theoretically be able to provide intermediate results in some fashion, no prior work to our knowledge has provided a dataset that can be used to directly learn model portfolios that outperform state of the art AutoML systems. Further, our work is easily extensible with additional methods to increase the strength of the learned portfolios, as demonstrated in our rebuttal where we were able to quickly add four additional model families (TabPFN, FT-Transformer, KNeighbors, and Linear models) to achieve improved results.
> > >
> > > We would like to conclude by re-stating the impact of our work. By providing a large-scale dataset that can produce learned portfolios that outperform the strongest tabular prediction methods and that can be easily extended with new methods over time, we enable easy collaboration and comparison within the research community to identify the impact of new models, new ensembling strategies, and new meta-learning techniques in order to pick the best combination via portfolio simulation which pushes the state of the art in tabular prediction forward and can be readily incorporated into widely used AutoML systems (The learned portfolios are compatible with AutoGluon and the learned portfolios can be passed as input arguments to an AutoGluon fit call).

---

### Official Review · Reviewer_kTk5 · 2023-11-03

**Soundness:** 3 good
**Presentation:** 2 fair
**Contribution:** 2 fair
**Rating:** 5
**Confidence:** 3

**Summary:**

Authors introduce a large dataset of tabular model evaluations on a large set of models as well as datasets. The prediction outputs of the considered models are also provided for efficient analysis without having to reevaluate the models. Authors demonstrate the utility of their dataset by 1. comparing hpo methods and auto-ml systems, 2. demonstrating ensembling, portfolio-selection and 3. transfer learning capabilities.

**Strengths:**

- The evaluation is extensive, with all the models constructed through bagging on multiple cross-validation folds and initialization for each dataset.
- Utility of TabRepo is demonstrated by analyzing the cost of tuning and the performance obtained for various auto-ml methods.
- Model portfolio  construction and transfer learning is shown to be effective using already-computed predictions.

**Weaknesses:**

As a dataset of tabular-model evaluations, the work is sound. However, I am not entirely convinced about the utility of the analysis provided in this work. For ex, various autoML methods and their performance comparisons (Fig 2) are already provided as part of AutoGluon. It would be helpful if the authors could illustrate a few more cases which potentially could benefit from TabRepo.

**Questions:**

Could you suggest few more potential use-cases that benifit from including prediction outputs?

---

> ### Author Response · Authors · 2023-11-20
> **Answer**
>
> Thank you for your review.
>
> ```
> * As a dataset of tabular-model evaluations, the work is sound. However, I am not entirely convinced about the utility of the analysis provided in this work. For ex, various autoML methods and their performance comparisons (Fig 2) are already provided as part of AutoGluon. It would be helpful if the authors could illustrate a few more cases which potentially could benefit from TabRepo.
> * Could you suggest few more potential use-cases that benifit from including prediction outputs?
> ```
> We agree that Fig 2 is already provided as part of AutoGluon's original paper, however it is expensive to reproduce and evaluate ensembles of methods given it requires fitting them on a large collection of datasets. The key contribution of this paper is to make this process much cheaper, by just requiring loading model predictions from disk instead of fitting models.
>
> Regarding use-cases for TabRepo, we believe that the performance of the portfolio ensemble in section 5 shows an important result for the community, namely that a simple ensemble of models can beat all state-of-the-art current AutoML systems. We believe this is an important result as it leads to methods with much lower latency and cost.
>
> Regarding more potential use-cases, we see a lot of potential work as our dataset allow to use many methods while considering ensembling without requiring to fit any model. This is critical since ensembling is almost always required to reach state-of-the-art performance in AutoML systems. For instance potential use-cases could include tuning hyperparameters and models (CASH) or applying early-stopping/multi-fidelity techniques that evaluate configurations partially and only let the top ones run with larger budget.

---

### Author Response · Authors · 2023-11-20
**General answer to reviewers**

We thank all reviewers for their valuable feedback.

We want to highlight an important aspect of our paper: namely that the paper provides a dataset that allow to simulate any ensemble from a large set of datasets (200) and models (currently 1206).

The fact that we can evaluate any ensemble cheaply is the critical part of the contribution for this dataset. We illustrated one important use-case by showing that an ensemble built on top of portfolios outperforms or matches the performance of all current AutoML and tabular methods.

We believe this is an interesting finding in itself for the community but above all we think this illustrates the potential research directions, namely to be able to evaluate and build state-of-the-art methods at a fraction of the current cost. This can enable a lot of research to improve the state-of-the-art for tabular methods while considering ensembles, for instance considering multi-fidelity, multi-objective or other transfer learning techniques.

Two reviewers mentioned that adding some models (in particular deep-learning ones) could improve the dataset and the paper. We agree, and have evaluated 4 more model families:
* a linear model
* a nearest neighbor
* one modern parametric DL model (FT-Transformer from Gorishniy et al 2021)
* one modern non-parametric DL model (TabPFN from Hollmann et al 2022)

We report here the aggregate results for those methods:

| method                                |   normalized-error |   rank |   time fit (s) |   time infer (s) |
|:--------------------------------------|-------------------:|-------:|---------------:|-----------------:|
| Portfolio (ensemble) (4h)        |              0.362 |  174.6 |         6597.5 |            0.061 |
| AutoGluon best (4h)                   |              0.389 |  208.2 |         5583.1 |            0.062 |
| Portfolio (4h)                   |              0.437 |  236.6 |         6597.5 |            0.013 |
| CatBoost (default)                    |              0.586 |  341.2 |          456.8 |            0.002 |
| FT-Transformer (default)               |              0.69  |  532.1 |          567.4 |            0.003 |
| LightGBM (default)                    |              0.714 |  491.5 |           55.7 |            0.001 |
| XGBoost (default)                     |              0.734 |  522.2 |           75.1 |            0.002 |
| MLP (default)                         |              0.772 |  629.4 |           38.2 |            0.015 |
| TabPFN (default)                      |              0.837 |  731.9 |            3.8 |            0.016 |
| LinearModel (tuned + ensemble) (4h)   |              0.855 |  873.8 |          612.4 |            0.038 |
| LinearModel (tuned) (4h)              |              0.862 |  891.6 |          612.4 |            0.006 |
| ExtraTrees (default)                  |              0.883 |  788.6 |            3   |            0     |
| RandomForest (default)                |              0.887 |  773.9 |           13.8 |            0     |
| LinearModel (default)                 |              0.899 |  940.1 |            7.1 |            0.014 |
| KNeighbors (tuned + ensemble) (4h)    |              0.928 |  980.8 |           12   |            0.001 |
| KNeighbors (tuned) (4h)               |              0.937 | 1016.5 |           12   |            0     |
| KNeighbors (default)                  |              0.973 | 1149.1 |            0.6 |            0     |

We ran only a single configuration for FT-transformer due to the large training cost on a GPU machine. We also ran a single configuration for TabPFN. In case of failures (which happens on ~30% of datasets due to current limitations on number of features and rows), we impute the result with a simple baseline than always run under 5 seconds.

As one can see, the FT-Transformer method performs in-between MLPs and the best boosted trees methods. Regarding TabPFN, the method does not reach the performance of top methods yet which is due to high failure rates from current method limitations (max 10 classes, max 100 features) and also due to the method not being able to currently exploit large numbers of rows effectively.

We added an appendix section with those additional results, see appendix F). We did not incorporate those results in the main section as 1) all models from this list (except linear) fail in non negligible amount of datasets due to algorithmic or hardware issues 2) FT-transformer requires an additional GPU as opposed to the model considered in the main sections (we are currently experimenting with CPU runs). That being said, those evaluations will of-course be available in the dataset.

We thank you again for your efforts and look forward to our discussion.

---

### Meta-Review · Area_Chair_btwR · 2023-12-15

**Metareview:**

One review is rather brief and superficial, but other reviews are of good quality and from experts. On balance, they say that this paper is below the high ICLR threshold. The paper provides a large dataset of tabular model metrics and predictions, which is a positive contribution, but reviewers feel that more depth is needed for ICLR.

**Justification For Why Not Higher Score:**

Reviewers have detailed and valid criticisms.

**Justification For Why Not Lower Score:**

The paper does represent substantial hard work.

---

### Decision · Program_Chairs · 2024-01-16

Reject